


# Future Rime Ice Conditions for Energy Infrastructure over Fennoscandia Resolved with a High-Resolution Regional Climate Model

Oskari Rockas[1], Pia Isolähteenmäki[1], Marko Laine[1], Anders V. Lindfors[1], Karoliina Hämäläinen[1], and Anton Laakso[1]

[1]Finnish Meteorological Institute, Erik Palménin aukio 1, 00560, Helsinki, Finland

**Correspondence:** Oskari Rockas (oskari.rockas@fmi.fi)

**Abstract.** Societies today are increasingly reliant on electricity, underscoring the need for reliable energy production. In cold climate regions, ice accumulation can cause significant harm to structures such as power transmission lines, leading to power loss or, in the worst case, the collapse of wires or transmission towers. Thus, since climate change is expected to impact winter weather conditions in northern Europe, its effects on atmospheric icing occurrence over the Fennoscandian region is a

crucial area of study. Here we utilize an ice accretion model based on ISO 12494, driven by outputs from the high-resolution regional climate model HCLIM, to analyze in-cloud icing conditions over two twenty-year periods: mid-century (2040–2060) and end-of-century (2080–2100). The regional outputs are bounded by two global climate models (EC-EARTH and GFDL-CM3, respectively) under the highly warming RCP 8.5 emission scenario. The results suggest a general decrease in in-cloud icing conditions over northern Europe compared to the historical period (1985–2005). An exception lies in the northern parts

of Fennoscandia and locally over higher altitudes, where some increasing trend is seen, particularly for annual maxima. Under the RCP 8.5 scenario, freezing temperatures become less common; however, rising temperatures allow for more moisture, potentially enhancing in-cloud icing if enough freezing temperatures remain.

## 1  Introduction

Atmospheric icing poses significant challenges for various sectors in cold climate regions globally. Energy infrastructure, in

particular, is vulnerable to disruptions due to ice accumulation on power transmission lines or wind turbines (Hämäläinen and Niemelä, 2017; Zhou et al., 2011; Chang et al., 2007; Frick and Wernli, 2012). For wind turbines, ice on the blades disturbs power production and accelerates blade wear, while transmission lines can experience problems such as galloping, and consequently, wire and tower breakage due to the combined effect of strong winds and icing (Chen et al., 2022; Havard and Van Dyke, 2005). Moreover, icing can reduce the efficiency of energy transmission by contributing to a phenomenon

known as corona loss (Sollerkvist et al., 2007; Yin et al., 2017; Lahti et al., 1997), in which the air surrounding a conductor is ionized and a corona discharge occurs.

Atmospheric icing has been studied for nearly a century (Arenberg, 1943; Lewis et al., 1947). It is an umbrella term that encompasses multiple types of ice accretion: rime ice due to supercooled liquid cloud droplets, glaze ice forming primarily by



accretion of freezing rain or drizzle, and wet snow icing due to adhesion of wet snow or sleet just above freezing conditions
(Makkonen, 2000). Suitable conditions for ice accretion depend on several meteorological parameters such as air temperature,
cloud liquid water content, wind, and precipitation. Climate change is predicted to alter their distribution in mid-latitudes,
particularly during the winter season (Lind et al., 2022); increases in temperature, moisture, and precipitation are expected.
Thus, changes in the occurrence of atmospheric icing are anticipated in Finland and the broader Fennoscandian region, as has
been discussed in previous research (Lutz et al., 2019; Iversen et al., 2023).

Both Lutz et al. (2019) and Iversen et al. (2023) examined how rime ice conditions - and in Iversen et al. (2023)'s case, also
wet snow conditions - are projected to evolve from the perspective of power transmission lines. Iversen et al. (2023) utilized the
regional weather model known as WRF (Weather Research and Forecasting) with a horizontal resolution of 12 km to downscale
two climate model configurations, furthermore assessing the changes in rime ice and wet snow loads. For rime ice, the largest
changes were predicted over the Scandinavian mountains, with differing trends between the two configurations; over western
Finland, a mostly decreasing trend was noted. Iversen et al. (2023) recommended an ensemble approach for planning future
ice load designs.

Lutz et al. (2019) employed an ensemble approach, using the output of 11 regional EURO-CORDEX climate simulations
with 12 km resolution to assess the change in icing on power lines. The simulations were combinations of six global climate
models (GCM) and three regional climate models (RCM). They found predominantly decreasing trends in northern Finland,
with mixed signals for southern and central Finland. They concluded that the regional climate model had more effect on
the results than the global climate model through the influence of cloud liquid water on icing. However, cloud liquid water
content was not included in the EURO-CORDEX simulations, and therefore they had to estimate it from relative humidity
(for cloudiness), specific humidity (water content), and temperature (fraction of liquid water). Other studies on the impacts of
climate change on icing conditions have focused mainly on cases related to wet snow or freezing rain (Kämäräinen et al., 2018;
Rácz et al., 2022; Marinier et al., 2023).

In this paper, we utilize outputs from a high-resolution regional climate model HARMONIE-Climate (HCLIM), with bound-
aries from two global climate models, to assess changes in in-cloud icing over the Fennoscandian region. The research has been
conducted under an EU HORIZON project called RISKADAPT (Asset Level Modelling of Risks in the Face of Climate In-
duced Extreme Events and Adaptation) with a focus on power transmission lines, similarly to previous research on the topic.
However, we can simulate atmospheric icing at multiple lower tropospheric heights (50–400 m) which also allows us to assess
the risks of in-cloud icing for wind production. In addition, we can simulate icing at a higher horizontal resolution than before
(3 km vs. 12 km in both Iversen et al. (2023) and Lutz et al. (2019)). It is important to note that a high horizontal resolution
comes with the expense of small ensemble size.

The paper is divided as follows: Section 2 details the icing model, climate model outputs, and icing data processing. Sec-
tion 3 presents the main results, including an overview of changes in mean and annual maximum ice loads from the entire
Fennoscandian area. In addition, changes in input parameters such as temperature are inspected to better discuss the results
and present conclusions in Section 4.



## 2 Data, models and methods

### 2.1 Rime ice model

Based on the icing algorithm of Makkonen (2000), a rime ice model has been developed at the Finnish Meteorological Institute and has been used, for example, to create an icing atlas for wind park planning (Hämäläinen and Niemelä, 2017). Rime ice accretion is modeled over a vertical (length = 1 m), freely rotating standard cylinder with icing rate $(\text{gs}^{-1})$ as its main output:

$$\frac{dm}{dt} = \alpha_1 \alpha_2 \alpha_3 w \mathbf{v} A \tag{1}$$

where $\alpha_1$ is the collision coefficient, $\alpha_2$ the sticking coefficient and $\alpha_3$ the accretion coefficient, while $w$ stands for the liquid water content $(\text{gm}^{-3})$, $\mathbf{v}$ denotes wind speed and $A$ the surface area of the cylinder. In other words, the algorithm represents how well the cloud droplets $(w)$ carried by the air stream $(\mathbf{v})$ aggregate $(\alpha_1, \alpha_2$ and $\alpha_3)$ to the cylinder $(A)$.

Drag and inertia control how well the particles in the air stream hit an object. This is represented by the collision coefficient $\alpha_1$, which is close to 1 for large particles (for which inertia dominates) and vice versa for small particles. The size of the particles depends on the median volume diameter (MVD), which itself is dependent on the liquid water content and the cloud droplet number concentration Nd. In this study, Nd was set as constant, $100 \text{ cm}^{-3}$ as reasoned by Hämäläinen and Niemelä (2017).

The sticking coefficient $\alpha_2$ expresses how efficiently air particles freeze upon contact with an object. In cases of rime ice, we are interested in supercooled water droplets that freeze immediately without rebounding; the same applies approximately for cases with rain droplets, so $\alpha_2$ is set to 1. The accretion coefficient $\alpha_3$ differs for dry growth cases (where $\alpha_3$=1) and wet growth cases (where there is a liquid layer on top of the ice surface; $\alpha_3 < 1$). ISO 12494 (International Organization for Standardization, 2017) explains in more detail how wet growth cases are managed in accordance with the accretion coefficient. The melting process in the ice model begins when the air temperature rises above 0°C.

From icing rate, various parameters related to icing are calculated; most notably the ice mass $(\text{gm}^{-1})$, but also the ice density $(\text{gm}^{-3})$, the total diameter of the cylinder and ice (Makkonen Lasse, 1984; Makkonen and Stallbrass J.R., 1984).

### 2.2 HARMONIE-Climate

Our study aims to investigate future icing conditions in the Fennoscandian region; thus, regional climate model projections are required as input for the rime ice model. HARMONIE-Climate (HCLIM) was chosen for this purpose because it provides all the necessary parameters to model rime ice in a high horizontal resolution.

HCLIM is based on the numerical weather prediction (NWP) model configuration of the ALADIN-HIRLAM NWP modeling system (Belušić et al., 2020; Lindstedt et al., 2015). Experiments by the Nordic Convection Permitting Climate Projections project (NorCP, Lind et al. (2020)) have been conducted with the HCLIM regional climate model in northern Europe using HARMONIE-Climate cycle 38 in two configurations: a lower resolution configuration (12 km) utilizing the ALADIN physics package (Termonia et al., 2018) and a higher resolution configuration (3 km) where deep convection is resolved (Bengtsson et al., 2017). Data from the latter were used in this study; the required parameters were calculated for multiple lower troposphere



levels (50 m, 100 m, 200 m, 300 m, and 400 m) with a horizontal resolution of 3 km and a temporal resolution of 3 hours. High horizontal resolution has its drawbacks, as it limits the ensemble approach due to the high computational costs of running the RCM.

RCMs require information on lateral boundary conditions, which are provided by large-scale global climate models. The HCLIM output we used was based on two boundary GCMs from the Coupled Model Intercomparison Project-Phase 5 (Taylor
et al., 2012), EC-EARTH and GFDL-CM3, chosen specifically to capture different types of climate response (Lind et al., 2022). EC-EARTH shows a colder and drier response to climate change in northern Europe compared to GFDL-CM3.

HCLIM output had been generated for three time periods spanning approximately 60 years in total; a control run from the historical period (1985–2005), a mid-century run (2040–2060), and an end-of-century run (2080–2100). These are referred to as historical, mid-century, and end-of-century, respectively hereafter.

In this study, we used the precalculated HCLIM dataset with Representative Concentration Pathway (RCP) emission scenarios RCP 4.5 (moderately warming) and RCP 8.5 (strongly warming). Since RCP 4.5 data was available only with one boundary model (EC-EARTH), scenario RCP 8.5 was selected allowing the use of two climate model datasets with different lateral boundaries (EC-EARTH and GFDL-CM3). The RCP 8.5 concentration pathway (business as usual) assumes continuously rising greenhouse gas emissions throughout the 21st century. Although RCP 8.5 is considered to be increasingly unlikely
due to the overestimation of coal use by the end of the century (Freistetter et al., 2022), it remains to be an useful tool for mid-century policy assessments (Schwalm et al., 2020).

## 2.3 Processing of the ice model outputs

The icing model output was further processed to allow a comprehensive evaluation of the changes in icing climate from multiple perspectives. The results can be divided into two categories: 1) results calculated for 50 meters to investigate icing on power
lines, and 2) results calculated for multiple lower tropospheric heights (50-400 m) relevant for wind energy production. The altitude of 50 meters was chosen for power lines as it was the lowest height above ground level where HCLIM output data were available. Table 1 lists the evaluated parameters and details the domains and heights they were analyzed at. In addition, we focused only on the months when freezing temperatures are common in northern Europe (October to April) in our analysis, following the example of Iversen et al. (2023).

Figure 1 displays the specific areas referenced in Table 1. Each area is named according to the municipality or one of the municipalities they fall under. In areas 1–5, which were selected to coincide with existing or planned wind parks, results were calculated from all heights. For power line perspective, separate areas were selected (areas 6–7) and in these regions only results at 50 m height were considered. In areas 6–7, a maximum of all grid points was calculated for each time step and further parameters were derived from that, including annual mean and maximum ice loads and icing episode durations.

For areas 1–5, extreme values of ice mass were calculated by taking the 99th quantile of the 20-year periods. To obtain these values, it was assumed that the climate is similar over a 20-year period; similar assumptions were made with the whole domain when comparing the changes across three distinct time periods. The icing hours and episode durations were calculated from the average of the grid points in the area. Here, it is also essential to emphasize that the definitions for an icing hour and an





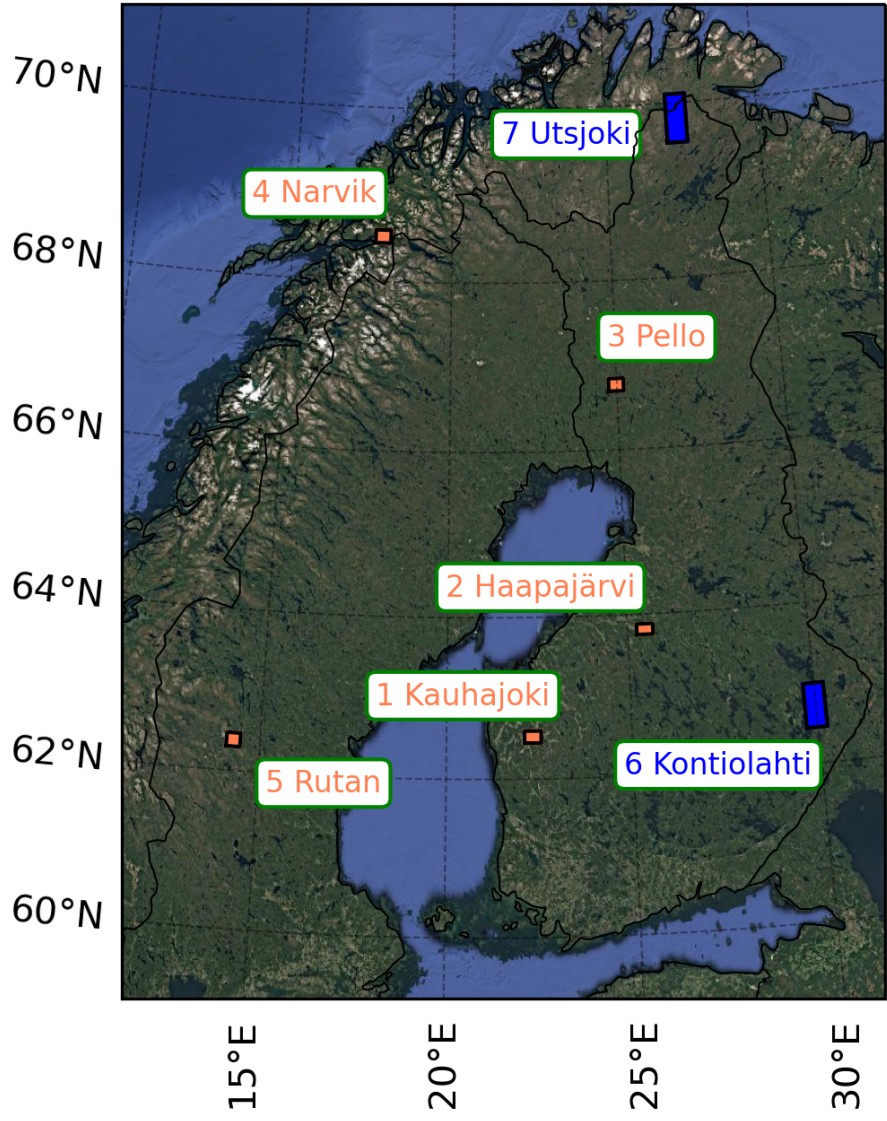

**Figure 1.** Map showing the locations to which specific results have been calculated. Marked with orange, areas 1–5 are regions where all heights have been considered and they align with existing or planned wind parks. Marked with blue, areas 6–7 are regions where results were calculated only at 50 meters with a focus on power transmission lines. (Map from © Google Maps.)

icing episode are different for areas 1–5 and 6–7. In areas 1–5, the presence of at least $10 \, \mathrm{g/m}$ of ice in the model cylinder is used as the definition for an icing hour. This applies to the icing conditions of a wind turbine (Hämäläinen and Niemelä, 2017; Turkia et al., 2013). For results calculated only at $50 \, \mathrm{m}$, icing hour is defined as an hour where the ice cover thickness $\geq 2\mathrm{mm}$. This limit is based on earlier research (Lahti et al., 1997) relating to corona losses caused by icing.



**Table 1.** Post-processed parameters and the domains to which they have been calculated are listed here. The locations of areas 1-7 can be found in Fig. 1.

| | Parameters | Entire domain (50m) | Areas 6-7 (50m) | Areas 1-5 (50-400m) |
|---|---|---|---|---|
| 1. | Mean/maximum ice mass (kg/m) | x | x | |
| 2. | 99th quantile of ice mass (kg/m) | | | x |
| 3. | Maximum ice thickness (mm) | | x | |
| 4. | Icing hours, 2mm (h) | x | x | |
| 5. | Icing hours, 10g (h) | | | x |
| 6. | Icing episode duration, 2mm (days) | | x | |
| 7. | Icing episode duration, 10g (days) | | | x |

After computing the icing model with HCLIM data, issues considering the lower boundary of sea surface temperatures (SST) and sea ice concentrations in HCLIM data over multiple areas of the Baltic Sea for both GCMs were found. The errors were clear during winter (December-February) and reflected in the surface temperatures and precipitation fields in the same areas. Through discussion with NorCP data providers (Wang et al., 2024), we were informed that the error should have little influence on land output, as also indicated by previously published studies based on the same data (Lind et al., 2022; Freistetter et al., 2022; Dyrrdal et al., 2023); thus, results are shown only for land areas.

## 3 Results

### 3.1 Future changes in atmospheric rime ice conditions

The results concerning future changes in atmospheric rime ice conditions are divided into three sections. The first section provides a general overview of the changes in icing conditions across the entire study area, Fennoscandia, at the height of 50 meters. Changes in ice load (mean and annual maximum) and the average annual number of icing hours are studied. Sea areas are excluded from the statistics, as reasoned in Section 2.3. The second part focuses on two test study areas in Finland from the power line perspective. In the third section, results related to icing hour amounts are presented from the perspective of wind power production as results are considered from five output heights between 50 and 400 meters.

#### 3.1.1 Fennoscandia (50 m)

For the entire study area, changes in both the ice mass and the hours of ice over the standard cylinder are presented for the height of 50 meters. Figure 2 illustrates the projected relative changes in the mean and annual maximum ice loads. A generally decreasing trend in mean ice loads is evident in Fig. 2a. However, over northern Lapland, northern Norway and





Kola-Peninsula, both models simulate locally a 30–50, even close to an 80-100 % increase during mid-century. By the end-of-century, a decreasing trend covers almost the entire area, and the strongest negative trend is obtained with the GFDL-CM3 boundary model. The relative trend is pronounced, with some regions experiencing close to a 100 % decrease, indicating conditions where icing would no longer occur. However, the mean loads in these regions are relatively low to begin with, resulting in a small absolute change, as seen in Fig. A1a.

There is greater variability in the results for maximum ice loads. Figure 2b depicts the relative change in annual maximum ice loads. Although the increasing trend in mean ice loads in northern Lapland and Norway becomes neutral or slightly decreasing toward end-of-century, a similar increasing trend persists until end-of-century for annual maxima. The increasing trend is more pronounced for EC-EARTH, where the increase is greater than 50% in northern Lapland and Norway for both time spans; in GFDL-CM3, the increasing trend diminishes by end-of-century. In addition, local increases of 20–50 % are observed elsewhere over northern Fennoscandia and the Scandinavian mountains in both outputs, especially in the mid-century period. Some local increases can be seen in southern parts of Finland and Sweden, but as was the case with mean ice loads, the absolute changes

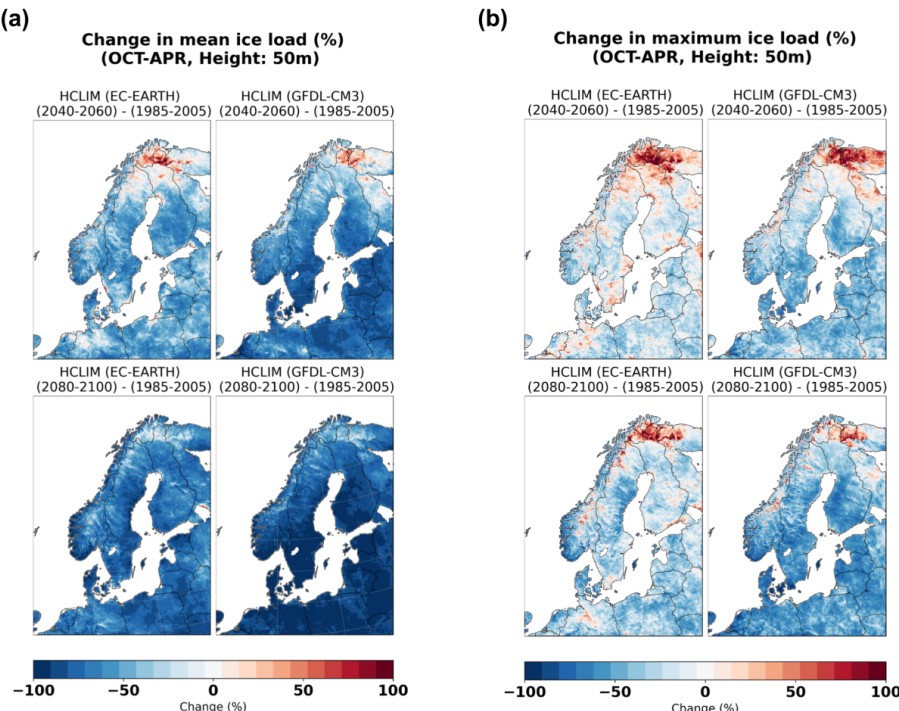

**Figure 2.** Relative change in a) mean ice load, b) mean annual maximum ice load at the model height of 50 meters. In the upper panel, the changes in icing from the historical period to mid-century and for both model configurations (EC-EARTH and GFDL-CM3) are shown. In the lower panel, the change from the historical period to end-of-century is shown.





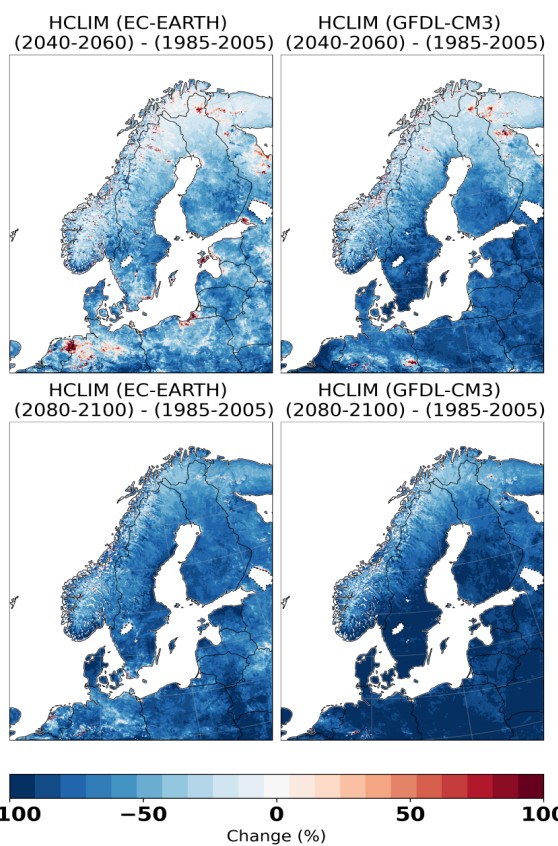

**Figure 3.** Relative change in annual mean icing hours at the model height of 50 meters. In the upper panel, the changes in icing from the historical period to mid-century and for both model configurations (EC-EARTH and GFDL-CM3) are shown. In the lower panel, the changes from the historical period to end-of-century are shown.

are small in these regions, as indicated in Fig. A1b. However, the decreasing trend over western Finland in GFDL-CM3 is also evident for absolute values (locally above -0.1 $kg/m$ and 50 to 100 % in Fig. 2b).

160     Beyond the actual ice masses accumulated on structures, the duration of the presence of ice is crucial; hence, changes in icing hours were examined. Here, as explained in Section 2.3, an icing hour is described as an hour with an ice cover thickness of $\geq 2mm$ in the model cylinder. Figure 3 visualizes the relative change in annual mean icing hours in the Fennoscandian area. A decreasing trend is prevalent in most of the study area, with a more substantial decrease in GFDL-CM3-based outputs; in southern and central Finland, a decrease of 80–100 % is projected by the end-of-century. For absolute changes in icing hours

165 (Fig. A2), the largest changes are in northern parts of Finland, central and northern Sweden, and the Scandinavian mountains, where a decrease of 500–1000 hours annually (20–40 days) is projected for the mid-century and close to 1000 hours or even





more for the end-of-century (40 days or more). This corresponds to a 10–50 % decrease in the mid-century; by the end-of-century, a 60–80 % decrease is simulated over central Sweden and southern Lapland, while northern Lapland is expected to see a 40–60 % decrease. However, locally in mid-century, especially in the northeastern parts of the domain, the model projects some increase.

### 3.1.2 Test areas 6–7: Power line perspective (50m)

To study changes in icing conditions in more detail, various areas were selected for further analysis. This section presents results related to ice loads, thicknesses, and icing episode durations at 50 meters from two areas presented in Fig. 1; one in eastern Finland (6: Kontiolahti) and the second in northernmost Finland (7: Utsjoki). Additionally, in Section 2.3 we have outlined the methods and definitions used to derive the parameters applied here, such as an icing episode.

For the Kontiolahti area, the results are presented in Fig. 4a–b, while Fig. 4c–d shows the same for Utsjoki. In box plots (Fig. 4a and Fig. 4c), the evolution of regional maximum ice loads are visualized, with solid lines denoting HCLIM with EC-EARTH and dashed line denoting HCLIM with GFDL-CM3. The white lines inside the boxes indicate the median value, the boxes themselves cover data from the first to the third quantile, and the whiskers extend from the middle 50% close to the extremes of the distribution. Both the annual mean and the annual maximum are calculated for the regional maximum of the grid points. A decreasing trend is evident for the yearly means (blue boxes), particularly in Kontiolahti, but also towards the end-of-century in Utsjoki. The predicted changes in the maxima of the mean distribution according to the GFDL-CM3 boundary are noticeable, as they become smaller than the median value in the historical period for both locations. With Utsjoki, the results exhibit greater variability as EC-EARTH shows a small increase in the mid-century, and in GFDL-CM3 the extreme values are larger in the mid-century than in the historical period.

With yearly maxima (yellow boxes), the signal is less clear compared to the annual means. Overall, in Kontiolahti, the median values decrease slightly when comparing future periods with the historical period; between future periods, the trend is neutral. In Utsjoki, both model configurations show increase in mid-century and, for EC-EARTH, the maxima of the distributions increase until the end-of-century. Thus, while mean ice loads are expected to mostly decrease (as discussed in Section 3.1.1), the potential for larger ice loads remains.

Changes in the relationship between the maximum thickness of the ice cover during an icing episode and the duration of such an episode are depicted for both model configurations in Figures 4b and 4d. Yellow represents the historical period, red the mid-century, and black the end-of-century. Triangles denote HCLIM with EC-EARTH, and circles HCLIM with GFDL-CM3. There are some regional differences in the scatter plots; for Kontiolahti (Fig. 4b), a shift toward shorter episodes and thinner ice covers is expected, while in Utsjoki, thick covers persist even with decreasing episode lengths (Fig. 4d). In Kontiolahti, icing episode lengths of 0–40 days are most common with a maximum thickness of 3–15 mm, with the largest thicknesses of 20–40 mm occurring when icing episode lengths exceed 20 days. HCLIM with EC-EARTH suggests more potential for thick covers (20–40 mm) and longer episodes (over 40 days) than HCLIM with GFDL-CM3. In Utsjoki, maximum thicknesses are generally around 40–60 mm across boundary models and time periods. In mid-century, there is indication of an increased chance of thicknesses of more than 60 mm, especially in HCLIM with GFDL-CM3.



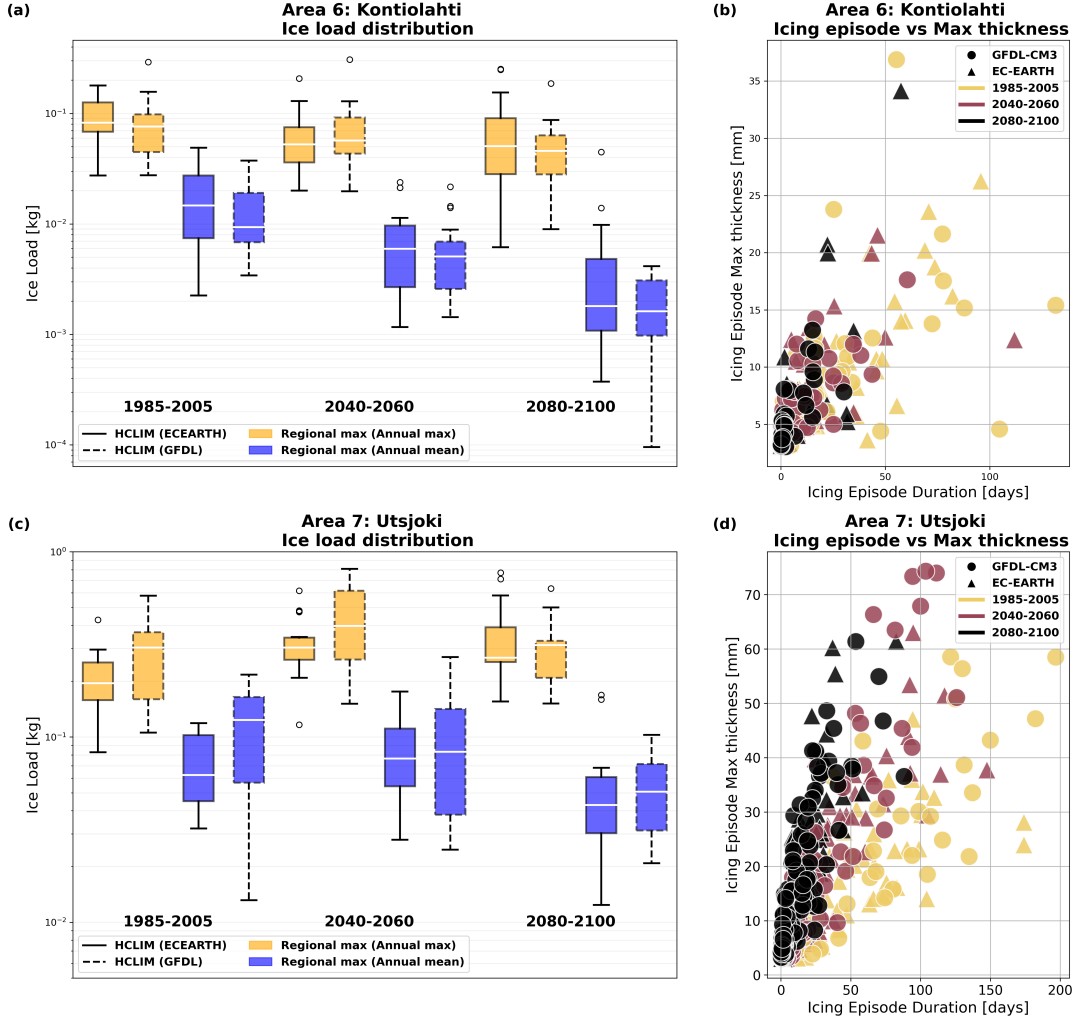

**Figure 4.** Regional results at the height of 50 meters from Kontiolahti (a-b) and from Utsjoki (c-d). In a) and c) distributions of yearly maxima (yellow) and means (blue) are shown for the regional maxima. Boxes with solid lines are EC-EARTH and dashed lines GFDL-CM3. The white line indicates the median value while the boxes cover the interquartile range. Whiskers extend close to the extremes of the distributions, and dots cover the outliers. In b) and d) the relation of icing episode duration and maximum thickness of ice cover during the episode are shown. Triangles denote EC-EARTH and circles GFDL-CM3 while historical values are shown in yellow, mid-century in red and end-of-century in black. Icing episode denotes to the period when ice cover thickness $\geq 2$mm.





### 3.1.3 Test areas 1-5: Wind power perspective (50-400m)

In this section, results from all five output heights (50, 100, 200, 300 and 400 m) are investigated in the case study area of Kauhajoki located in western Finland along with four additional areas around Fennoscandia. These areas correspond to areas 1–5 presented in Fig. 1, and they coincide with existing or planned wind park locations. Consequently, here an icing hour is

described differently from Section 3.1 to better align with wind power standards ($\geq 10$ g/m of ice present in the cylinder). However, an important factor here is to note that the model output was calculated for a cylinder of diameter of 3 cm, which is more suited for transmission lines than for wind turbines.

For Kauhajoki, yearly icing hours at heights 50–400 m are presented in Fig. 5, with distributions presented for each time period. The mean value is highlighted with a diamond symbol. It is evident that the number of icing hours decreases with time

at each height and for both boundary models. For HCLIM with GFDL-CM3 (Fig. 5b), at upper heights of 200–400 meters, the mean icing hour values in the mid-century period align with or are smaller than the minimum values in the historical period, while at the end-of-century, the simulated maximum hours align with or are smaller than the minimum values in history. With EC-EARTH (Fig. 5a), the distributions in future periods cover a wider spectrum, although with a similar decreasing trend. At

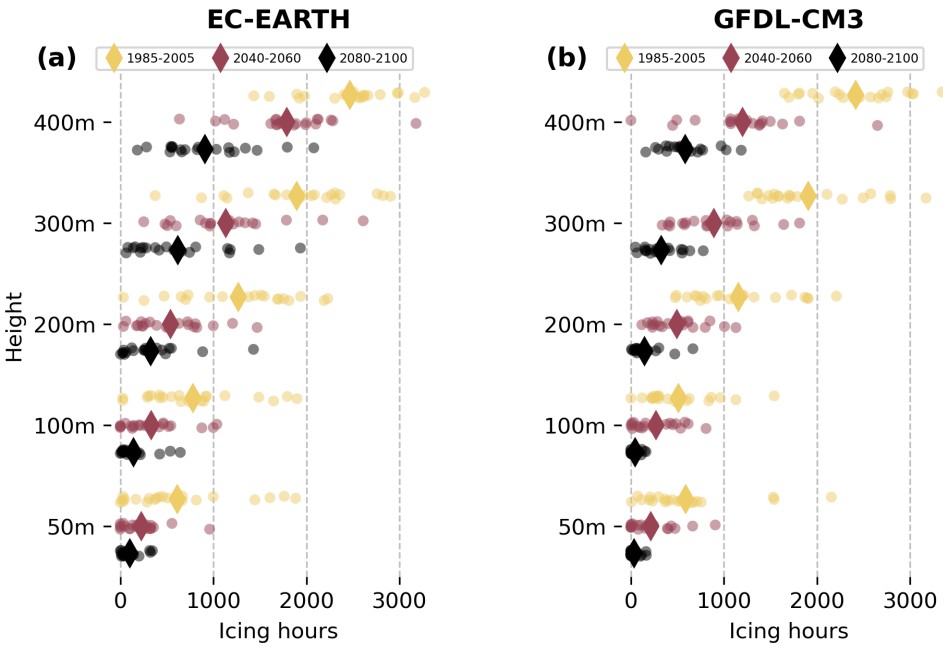

**Figure 5.** The distribution of annual icing hours at different heights (ranging from 50 m to 400 m) averaged over a box in area 1 (Kauhajoki) with a) EC-EARTH b) GFDL-CM3 as the boundary climate model. Yellow dots denote the historical period, red the mid-century and black the end-of-century with vertical line indicating the conditional mean value.





lower heights of 50–100 m, there are individual years with more than 1000 hours of significant ice present in the historical
period, but none forecasted for the end-of-century. If we compare the heights of 300 and 100 m, the mid-century mean in 300
m is greater than the historical mean in 100 m for both configurations; an interesting point to consider, as wind turbine heights
have increased with time. A similar plot of the distributions of icing episode durations is presented in Fig. A5; the decrease in
durations for this location is strongest in HCLIM with GFDL-CM3.

In Table 2, results for Kauhajoki can be found in numerical format, including the maximum and minimum icing hours for
each height and time period. In addition, the maximum icing episode length (when averaged over the area) and the range of
the 99th quantile of ice loads (maximum-average-minimum in the area) for each period are presented. As noted before, with
icing hours and icing episodes, the trend is towards decreasing amounts; according to HCLIM (GFDL-CM3), in the future, a
year without icing could occur even at 400 m height in Kauhajoki. Also notable is the decrease in the maximum icing episode
lengths; from 50–100 days to around 5–40 days by the end-of-century depending on the boundary model and height. For the
ice masses, the signal is also decreasing, and with HCLIM (GFDL-CM3), the average 99th quantile is 0.1 $\mathrm{kg/m}$ even at 400
meters by the end-of-century.

In Tables A2–A4, results are collected from additional study areas. For Pello (area 3 in northwestern Finland, Table A2),
results indicate mixed signals for the 99th quantile ice mass at the height of 50 m with decreasing trends from 100 meters
upwards (from highest values of 1–3 $\mathrm{kg/m}$ in the area in the historical period to 0.4–2 $\mathrm{kg/m}$ in future periods). For the rest of
the areas, trends in both ice masses and icing durations are decreasing. In area 2 (Haapajärvi, Table A1), by the end-of-century,
the 99th quantiles become dominantly smaller than 1 $\mathrm{kg/m}$ across both models and all heights, while the maximum episode
durations change from 70–175 days to around 15–50 days. For area 4 (around Narvik, Norway; Table A3), the maxima of
the 99th quantile ice loads range from 20 to 40 $\mathrm{kg/m}$ in history and from 7 to 20 $\mathrm{kg/m}$ at end-of-century across heights and
both model configurations. For Rutan (area 5 in central Sweden, Table A4), maximum durations decrease from over 100 days
to under 100 days starting from mid-century, while the 99th quantile loads are reduced by more than half in HCLIM with
GFDL-CM3 (less with EC-EARTH).

## 3.2   Future changes in HCLIM output parameters

The meteorological input parameters from HCLIM to the rime ice model are temperature, cloud liquid water content, and
wind speed. Hence, changes in icing are a consequence of changes in these three parameters, and in this section, the simulated
changes of these parameters at the height of 50 meters are investigated separately.

### 3.2.1   Temperature

Ice formation takes place at freezing temperatures; therefore, it is important to consider how often freezing conditions occur.
The critical threshold temperature between icing and melting in the rime ice model is set to 0 °C, above which water remains
liquid or the existing ice melts. In reality, this can vary depending on the type of icing and other meteorological conditions, such
as the presence of ice nucleation particles (International Organization for Standardization, 2017). In contrast, at sufficiently low



**Table 2.** Numerical results for Kauhajoki (area 1, western Finland). CTRL is for the historical period (1985–2005), FUT1 is the mid-century (2040–2060) and FUT2 the end-of-century (2080–2100). For each time period and GCM (EC-EARTH and GFDL-CM3, respectively), six numerical values are given. IM (Max Avg Min) is for the regional maximum, average and minimum of the 99th quantile of ice mass. IE Max denotes the maximum ice episode duration taken from the average of the area and IH (Min & Max) denote the minimum and maximum icing hour amounts, taken also from the regional average.

| | | | EC-EARTH | | | | GFDL-CM3 | | | |
|---|---|---|---|---|---|---|---|---|---|---|
| **Area** | **Height** | **Year** | **IM** | **IE** | **IH** | **IH** | **IM** | **IE** | **IH** | **IH** |
| | | | Max Avg Min | Max | Min | Max | Max Avg Min | Max | Min | Max |
| 1. | 50m | CTRL | 0.07 0.06 0.04 | 53 | 0 | 1914 | 0.10 0.07 0.05 | 84 | 0 | 2160 |
| | | FUT1 | 0.05 0.04 0.03 | 37 | 0 | 1239 | 0.06 0.04 0.02 | 34 | 0 | 981 |
| | | FUT2 | 0.03 0.02 0.01 | 15 | 0 | 639 | 0.01 0.01 0.01 | 7 | 0 | 306 |
| | 100m | CTRL | 0.13 0.08 0.05 | 53 | 0 | 1992 | 0.16 0.12 0.09 | 47 | 0 | 1812 |
| | | FUT1 | 0.06 0.04 0.03 | 25 | 0 | 1605 | 0.09 0.05 0.04 | 34 | 0 | 939 |
| | | FUT2 | 0.04 0.03 0.02 | 25 | 0 | 771 | 0.02 0.01 0.01 | 7 | 0 | 546 |
| | 200m | CTRL | 0.34 0.24 0.11 | 80 | 6 | 2478 | 0.29 0.18 0.11 | 83 | 60 | 2727 |
| | | FUT1 | 0.15 0.08 0.05 | 33 | 3 | 1716 | 0.12 0.08 0.06 | 39 | 6 | 1962 |
| | | FUT2 | 0.13 0.08 0.04 | 34 | 0 | 1656 | 0.03 0.02 0.01 | 23 | 0 | 762 |
| | 300m | CTRL | 0.79 0.41 0.24 | 76 | 213 | 3090 | 0.48 0.27 0.16 | 97 | 507 | 3258 |
| | | FUT1 | 0.27 0.18 0.11 | 50 | 39 | 2706 | 0.35 0.15 0.09 | 48 | 0 | 2343 |
| | | FUT2 | 0.28 0.15 0.10 | 34 | 0 | 2064 | 0.07 0.05 0.03 | 22 | 0 | 993 |
| | 400m | CTRL | 2.28 1.37 0.58 | 83 | 870 | 3351 | 1.01 0.68 0.44 | 97 | 981 | 3405 |
| | | FUT1 | 0.51 0.33 0.22 | 50 | 225 | 3306 | 0.49 0.32 0.17 | 61 | 0 | 2748 |
| | | FUT2 | 0.39 0.26 0.18 | 35 | 57 | 2178 | 0.16 0.10 0.07 | 19 | 0 | 1266 |

temperatures, around and lower than -20°C, ice accumulation from supercooled liquid droplets is limited, as the air can only hold a small amount of liquid water (Westbrook and Illingworth, 2011).

Exploring changes in the occurrence of the critical temperature threshold is important, as it dictates the effect of temperature on icing. In Fig. A3, the mean temperatures of a typical icing season (October-April) at 50 m are shown for the historical
period and the two future periods (2040–2060, 2080–2100). The boundary model in the upper panel is EC-EARTH, and in the lower panel, GFDL-CM3. It is evident that the mean temperature in the historical period is widely below 0 degrees from the northern to central parts of the domain. By the end-of-century, the mean temperature is projectede to be below zero only in northern Lapland and over Scandinavian mountains corresponding to the strongly warming emission scenario RCP 8.5.

An important factor to consider is how much time is spent under freezing conditions even in a highly warming future. Figure
6 shows the average hours spent below zero degrees at the height of 50 m in historical and future periods. The time spent in freezing conditions decreases everywhere in the domain. Excluding southern Sweden and some of the coastal regions in Nor-



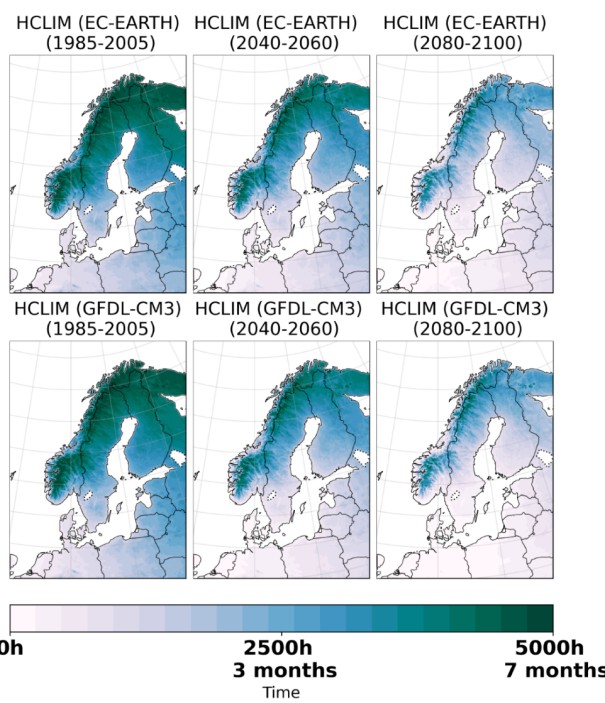

**Figure 6.** The average time that temperature is below zero at 50 m model height (October to April). The upper row shows the evolution of time spent below freezing for HCLIM with EC-EARTH, bottom row for HCLIM with GFDL-CM3. The left panel is for the historical period, the central one for the mid-century and the right panel for the end-of-century period.

way, Fennoscandia experiences around 4–7 months of freezing temperatures in the historical period. For the mid-century, this drops down to 2–6 months except for mountainous regions. However, for the end-of-century, southern and central Fennoscandia fall under an average of 0–2 months, northern parts to 2–4 months, and the highest altitudes on average to 4–6 months of

freezing temperatures. Some areas in the southern part of the domain, such as the south-western coast of Finland, southernmost Sweden, coastal regions of Norway and Baltic states and Denmark, may experience close to zero months of freezing conditions at the end-of-century.

### 3.2.2   Liquid water content

The amount of liquid water content (LWC) in the atmosphere defines how much ice is possible to accumulate if other me-

teorological conditions are favorable. In our case, LWC only included the cloud liquid water content (CLWC) because cloud rain water content (CRWC) was not available in the pre-calculated HCLIM data. Thus, the modeled atmospheric ice type is primarily rime ice.



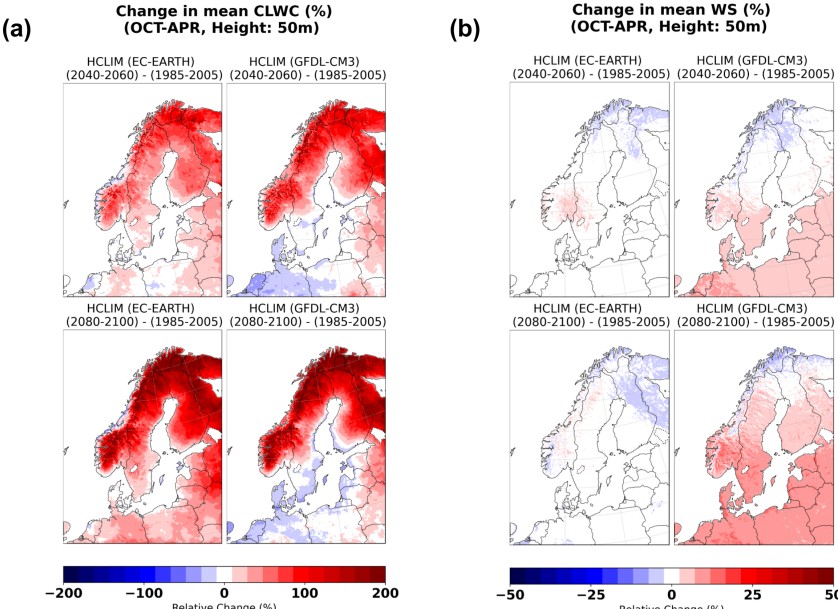

**Figure 7.** The relative change in a) mean cloud liquid water content, b) mean wind speed at model height of 50 meters for both GCMs (EC-EARTH in the left column, GFDL-CM3 in the right column). The upper rows compare the mid-century (2040–2060) with the historical period (1985–2005), while the bottom rows compare the end-of-century (2080–2100) with history.

The relative change in the mean CLWC values can be seen in Fig. 7a. An increasing trend (dominating red colors) is present in most of Fennoscandia for both mid-century and end-of-century, applying particularly for HCLIM with EC-EARTH. In the

southern part of the domain, GFDL-CM3 simulates a slight decreasing trend, by the end-of-century also in most of the Finnish and Swedish coastal areas. In EC-EARTH, a similar pattern is present on the western coast of Norway, while HCLIM with GFDL-CM3 predicts mostly an increasing trend. The absolute changes in the mean CLWC are shown in Figure A4a. The strongest absolute increase can be seen at higher altitudes, with an increase of $0.02 \ \mathrm{g/m^3}$.

### 3.2.3   Wind speed

The function of wind in relation to atmospheric icing is to transport liquid droplets to the surface of an object where the possible ice forms. If the wind speed is high, transportation is more efficient compared to situations with calmer winds. The icing model used in this study assumes a freely rotating cylinder, thus, wind direction is not considered.

In Fig. 7b, the relative changes in the mean wind speeds at the model height of 50 m are shown, while the changes in the absolute wind speeds are shown in Fig. A4b. Only minor changes in mean wind speeds are predicted, at least in HCLIM with

EC-EARTH. It has been noted in the research of Lind et al. (2022) that HCLIM (GFDL-CM3) simulates an increase in zonal flow during the autumn and winter seasons in Fennoscandia which aligns with the increase in the mean wind speed seen in Fig. 7b. In addition, a study by Ruosteenoja et al. (2019) comprehensively presented the effect of climate change on wind



conditions in northern Europe. They concluded that although the change in mean wind speed is minor, in autumn, the westerly wind speeds have a significantly increasing trend in northern Europe and in winter in northwest Europe.

## 4 Conclusions

The objective of this study was to assess the simulated changes in rime ice formation during in-cloud icing episodes across the Fennoscandian region. We utilized high-resolution climate model outputs driven by two GCMs (HCLIM with EC-EARTH and GFDL-CM3) under the highly warming RCP 8.5 scenario. In our study, the focus was both on a model height close to the surface (50 m) to investigate icing on power lines and on four additional heights (100–400 m) as atmospheric icing across the boundary layer impacts wind energy production in northern latitudes.

Both GCMs agree that, in general, there is a decreasing trend in mean rime ice conditions near the surface over most of Fennoscandia by the end-of-century. However, there is a temporary increasing trend during mid-century turning into a decreasing one towards the end-of-century over parts of northern Fennoscandia and at higher altitudes. The mid-century increase is approximately 30–50%, while the strongest negative changes approach 100% toward the end-of-century. Similarly, the time spent under icing conditions is predicted to generally decrease in the Fennoscandian region.

The estimated response of annual ice load maxima to climate change shows more variation than that of mean ice loads. While southern and central Fennoscandia are predicted to experience largely a decreasing trend, over parts of northern Fennoscandia and Scandinavian mountains, an increasing trend remains until the end-of-century. For the mid-century, the increasing trend covers most of northern Fennoscandia according to HCLIM with EC-EARTH, whereas with GFDL-CM3, the response is more restricted. In HCLIM with GFDL-CM3, a decrease in annual maxima in western parts of Finland is clear both relatively and absolutely; however, EC-EARTH indicates a less noticeable decrease. In addition, the relationship between the maximum thickness obtained during an icing episode and the episode duration was examined in two test areas. In eastern Finland, both the thickness and the duration are generally expected to decrease, while in northernmost Finland, large maxima are expected even by the end-of-century, but reached with shorter episode lengths.

For heights of 100–400 meters, the results primarily concern the frequency of significant icing for wind energy production. For example, in a test area in western Finland, few, if any, years are predicted to have more than 1000 hours (around 40 days) of icing by the end-of-century at heights below 200 m. In addition, only HCLIM with EC-EARTH suggests that the likelihood for years with more than 1000 hours remains at upper heights. However, when comparing icing on heights of 300 and 100 m, on average, more time is spent under icing conditions in mid-century in 300 m compared to 100 m in the historical period. Thus, as wind turbine heights increase, significant icing may occur at levels similar to or even greater than those observed today. Across other test areas and heights, the icing hours and episode lengths exhibit largely a decreasing trend.

Overall, in-cloud icing is expected to decrease by the end-of-century. This reduction is expected in both the amount of ice and the duration of icing periods. The predicted trends are stronger with HCLIM (GFDL-CM3), as of the two GCMs, it has a warmer response to climate change. We suggest that the main driver for the decrease is the warming trend in temperatures. Although the warmer atmosphere allows for a higher moisture content, icing does not occur when temperatures are above zero



degrees Celsius. The increasing trend over some of the northern regions in mid-century could be explained by the increase of LWC over regions where freezing temperatures remain, but, on the other hand, the temperatures are not too cold, allowing water to stay in liquid form. Correspondingly, this can explain the increase in the annual maxima for the end-of-century; even with an enhanced temperature increase, the likewise increased LWC allows for temporarily large ice loads.

Icing is a complex phenomenon due to a couple of reasons. Firstly, ice formation depends on numerous meteorological parameters, making its modeling, prediction, and observation challenging. For example, Lutz et al. (2019) noted in their research that the RCM had more control over ice amounts in the ensemble than the GCM through LWC and cloudiness. In our study, the absolute ice loads are smaller compared to the ensemble means in the study of Lutz et al. (2019) and thus may represent the lower end of the estimated ice distribution. This was supported by our rough comparisons with LWC obtained from the

CERRA reanalysis data (stands for Copernicus European Regional Reanalysis, (Ridal et al., 2024)). For 50 and 400 meters, about two times smaller amounts of mean LWC were observed in Finland in HCLIM data (not shown); however, for monthly maxima, HCLIM data showed larger amounts.

    Moreover, atmospheric icing encompasses various subtypes, each with distinct formation processes, which rely differently on influencing parameters such as ice density. The impact and severity of different icing types vary significantly depending on

the affected target: aviation and in-cloud icing, forests and wet-snow-damages, passive-icing episodes and wind turbines, and ice loads over power lines or electric towers.

    The RCP 8.5 pathway scenario used in this study is a business-as-usual scenario, sometimes called pessimistic, leading to substantial warming. Consequently, the predicted reduction in icing might be overestimated. The current understanding suggests that we are moving towards the RCP 4.5 scenario, but future decades could bring changes depending on the actions

taken. An additional uncertainty in our results is that they are based on the phase 5 Coupled Model Intercomparison Project (CMIP5)-projections which are suggested to have too little spread in their response to changes in the Atlantic meridional overturning circulation (AMOC) (Liu et al., 2017); the collapse of which would have major consequences in the regional climate of northern Europe (van Westen et al., 2024).

    Although our results indicate a future where icing events become rarer, atmospheric icing as a phenomenon will not disap-

pear, not even at the lowest near-surface layers. More research in high horizontal and vertical resolution is needed that uses an ensemble approach with both regional climate models and pathway scenarios. Subsequently, a better understanding of future icing conditions can be achieved to prepare and design the energy infrastructure to be sustainable.

*Data availability.* Datasets of the results of icing model simulations will be made available from FMI's Research Data repository.



## Appendix A: Figures

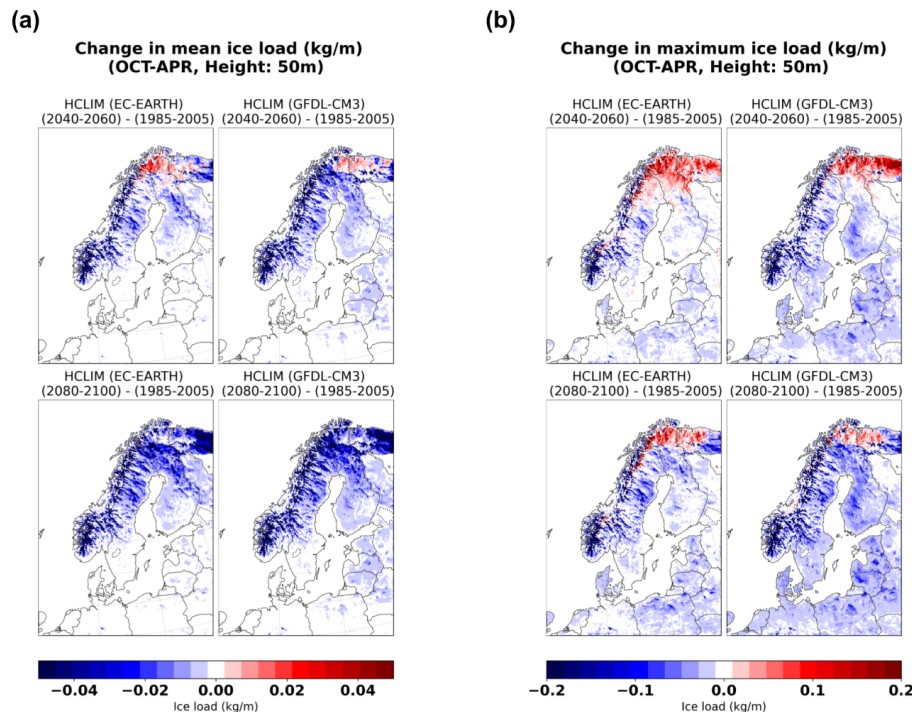

**Figure A1.** Same as in Fig. 2 but for absolute ice loads.





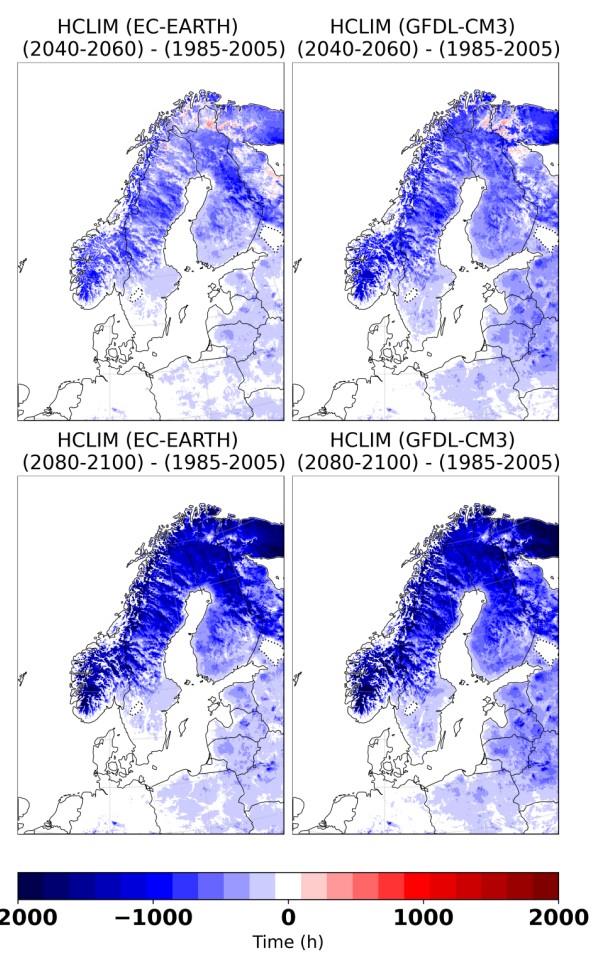

**Figure A2.** Same as in Fig. 3 but for absolute change in yearly icing hours.





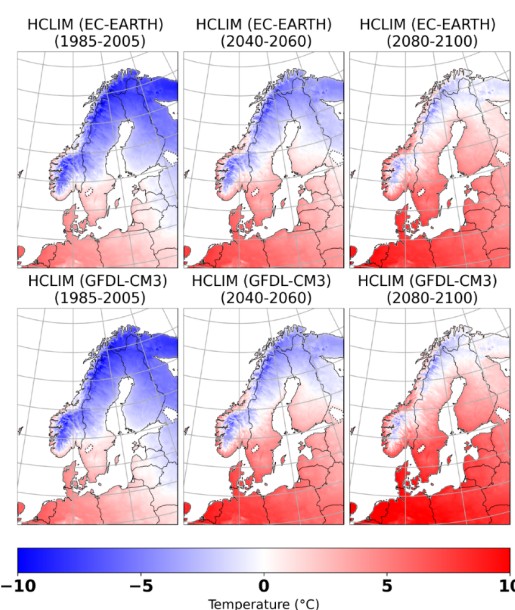

**Figure A3.** Mean temperature (50 m) of October-April -period in the historical (left panel), the mid-century (middle panel) and the end-of-century period (right panel) for HCLIM with EC-EARTH (upper row) and HCLIM with GFDL-CM3 (bottom row).



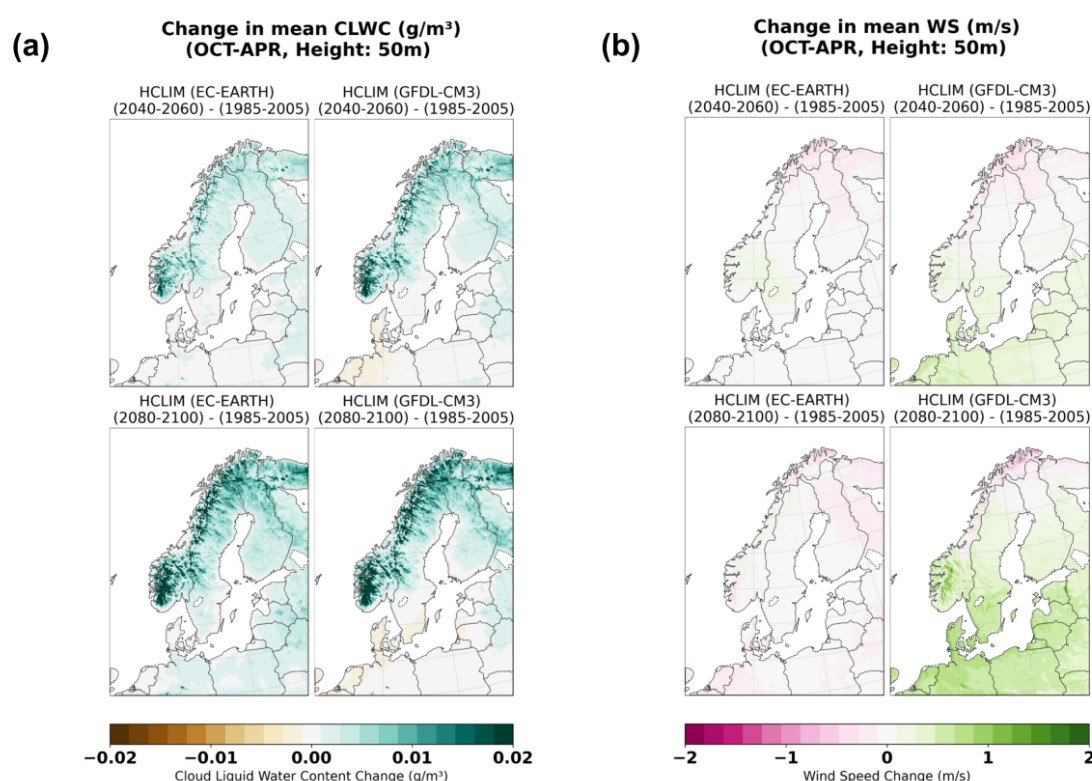

**Figure A4.** Same as in Fig. 7 but for absolute changes in a) mean liquid water content, b) mean wind speed at the height of 50 meters.





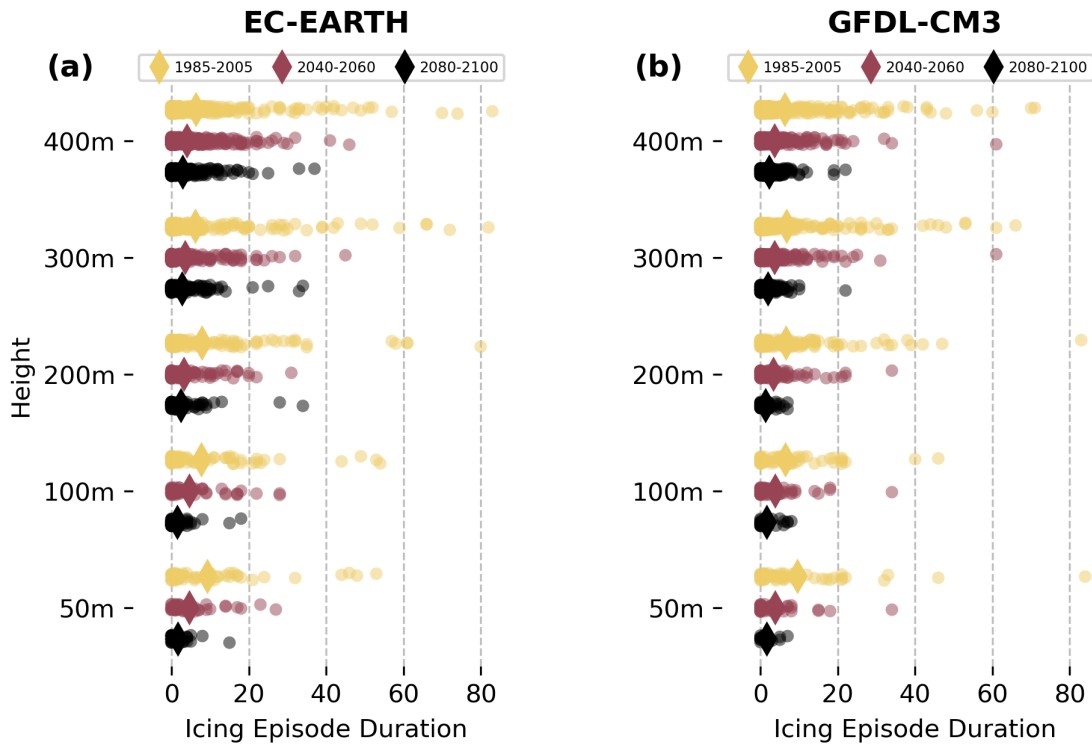

**Figure A5.** Same as in Fig. 5 but for icing episode durations at different heights (from 50 m to 400 m) with a) EC-EARTH b) GFDL-CM3 as the boundary climate model.





## Appendix B: Tables

**Table A1.** Numerical results for Haapajärvi (area 2, northern Finland). CTRL is for the historical period (1985–2005), FUT1 is the mid-century (2040–2060) and FUT2 the end-of-century (2080–2100). For each time period and GCM (EC-EARTH and GFDL-CM3, respectively), six numerical values are given. IM (Max Avg Min) is for the regional maximum, average and minimum of the 99th quantile of ice mass. IE Max denotes the maximum ice episode duration taken from the average of the area and IH (Min & Max) denote the minimum and maximum icing hour amounts, taken also from the regional average.

| | | | EC-EARTH | | | | GFDL-CM3 | | | |
|---|---|---|---|---|---|---|---|---|---|---|
| **Area** | **Height** | **Year** | **IM** | **IE** | **IH** | **IH** | **IM** | **IE** | **IH** | **IH** |
| | | | Max Avg Min | Max | Min | Max | Max Avg Min | Max | Min | Max |
| 2. | 50m | CTRL | 0.14 0.09 0.05 | 84 | 0 | 2373 | 0.21 0.14 0.08 | 72 | 0 | 1788 |
| | | FUT1 | 0.07 0.05 0.04 | 54 | 0 | 1881 | 0.11 0.05 0.02 | 34 | 0 | 1605 |
| | | FUT2 | 0.05 0.04 0.02 | 35 | 0 | 1569 | 0.02 0.02 0.01 | 16 | 0 | 630 |
| | 100m | CTRL | 0.28 0.19 0.08 | 112 | 0 | 2502 | 0.46 0.35 0.17 | 76 | 0 | 2511 |
| | | FUT1 | 0.12 0.09 0.06 | 43 | 0 | 1911 | 0.12 0.08 0.05 | 57 | 0 | 1656 |
| | | FUT2 | 0.12 0.08 0.05 | 36 | 0 | 2151 | 0.06 0.04 0.02 | 21 | 0 | 849 |
| | 200m | CTRL | 1.47 0.91 0.39 | 120 | 72 | 3480 | 0.99 0.78 0.45 | 156 | 0 | 3195 |
| | | FUT1 | 0.45 0.30 0.15 | 58 | 6 | 2712 | 0.31 0.22 0.14 | 62 | 108 | 2214 |
| | | FUT2 | 0.39 0.27 0.12 | 39 | 18 | 2508 | 0.15 0.11 0.07 | 28 | 18 | 1293 |
| | 300m | CTRL | 2.13 1.70 1.05 | 123 | 555 | 3690 | 1.13 0.98 0.78 | 156 | 1035 | 3651 |
| | | FUT1 | 0.96 0.61 0.42 | 78 | 216 | 3045 | 0.55 0.40 0.24 | 81 | 561 | 2322 |
| | | FUT2 | 0.36 0.29 0.18 | 39 | 63 | 2484 | 0.33 0.21 0.13 | 28 | 135 | 1713 |
| | 400m | CTRL | 2.30 1.97 1.26 | 122 | 1827 | 4020 | 2.05 1.28 0.86 | 174 | 1344 | 3981 |
| | | FUT1 | 1.50 0.99 0.59 | 81 | 879 | 3579 | 0.74 0.54 0.36 | 84 | 0 | 2646 |
| | | FUT2 | 0.65 0.49 0.37 | 47 | 363 | 2541 | 0.68 0.43 0.22 | 34 | 315 | 2139 |

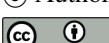



**Table A2.** Numerical results for Pello (area 3, Finnish Lapland). CTRL is for the historical period (1985–2005), FUT1 is the mid-century (2040–2060) and FUT2 the end-of-century (2080–2100). For each time period and GCM (EC-EARTH and GFDL-CM3, respectively), six numerical values are given. IM (Max Avg Min) is for the regional maximum, average and minimum of the 99th quantile of ice mass. IE Max denotes the maximum ice episode duration taken from the average of the area and IH (Min & Max) denote the minimum and maximum icing hour amounts, taken also from the regional average.

| | | | EC-EARTH | | | | GFDL-CM3 | | | |
|---|---|---|---|---|---|---|---|---|---|---|
| **Area** | **Height** | **Year** | **IM** | **IE** | **IH** | **IH** | **IM** | **IE** | **IH** | **IH** |
| | | | Max Avg Min | Max | Min | Max | Max Avg Min | Max | Min | Max |
| 3. | 50m | CTRL | 0.47 0.22 0.06 | 122 | 75 | 3783 | 0.51 0.20 0.07 | 157 | 165 | 3459 |
| | | FUT1 | 0.76 0.40 0.09 | 141 | 18 | 2958 | 0.42 0.22 0.09 | 94 | 117 | 2784 |
| | | FUT2 | 0.21 0.10 0.04 | 50 | 0 | 1965 | 0.25 0.12 0.05 | 64 | 6 | 2286 |
| | 100m | CTRL | 1.30 0.54 0.15 | 127 | 435 | 3864 | 1.08 0.49 0.15 | 162 | 0 | 3726 |
| | | FUT1 | 0.75 0.45 0.19 | 119 | 291 | 3438 | 0.75 0.41 0.18 | 96 | 351 | 3324 |
| | | FUT2 | 0.35 0.21 0.09 | 50 | 33 | 2277 | 0.40 0.23 0.12 | 64 | 51 | 2391 |
| | 200m | CTRL | 2.25 1.20 0.55 | 133 | 681 | 4263 | 1.61 1.04 0.52 | 178 | 1077 | 4128 |
| | | FUT1 | 1.15 0.80 0.42 | 120 | 591 | 3387 | 1.14 0.78 0.50 | 105 | 747 | 3441 |
| | | FUT2 | 0.75 0.40 0.20 | 51 | 138 | 2535 | 0.71 0.43 0.24 | 53 | 360 | 2454 |
| | 300m | CTRL | 2.24 1.47 0.76 | 146 | 1566 | 4308 | 2.25 1.50 0.72 | 188 | 1320 | 4140 |
| | | FUT1 | 1.70 1.08 0.57 | 122 | 1161 | 4242 | 1.47 1.10 0.69 | 106 | 1134 | 3531 |
| | | FUT2 | 1.11 0.63 0.33 | 51 | 261 | 2673 | 1.12 0.77 0.47 | 42 | 555 | 2916 |
| | 400m | CTRL | 3.14 2.17 1.11 | 146 | 1806 | 4437 | 2.94 2.16 0.99 | 148 | 1929 | 4464 |
| | | FUT1 | 2.45 1.56 0.94 | 122 | 1458 | 4449 | 2.45 1.53 0.95 | 106 | 0 | 3639 |
| | | FUT2 | 1.58 1.05 0.58 | 51 | 915 | 2697 | 1.45 1.08 0.71 | 45 | 528 | 3234 |



**Table A3.** Numerical results for Narvik (area 4, northern Norway). CTRL is for the historical period (1985–2005), FUT1 is the mid-century (2040–2060) and FUT2 the end-of-century (2080–2100). For each time period and GCM (EC-EARTH and GFDL-CM3, respectively), six numerical values are given. IM (Max Avg Min) is for the regional maximum, average and minimum of the 99th quantile of ice mass. IE Max denotes the maximum ice episode duration taken from the average of the area and IH (Min & Max) denote the minimum and maximum icing hour amounts, taken also from the regional average.

| | | | EC-EARTH | | | | GFDL-CM3 | | | |
|---|---|---|---|---|---|---|---|---|---|---|
| **Area** | **Height** | **Year** | **IM** | **IE** | **IH** | **IH** | **IM** | **IE** | **IH** | **IH** |
| | | | Max Avg Min | Max | Min | Max | Max Avg Min | Max | Min | Max |
| 4. | 50m | CTRL | 20.81 2.69 0.00 | 213 | 0 | 5112 | 25.02 3.00 0.00 | 213 | 0 | 5112 |
| | | FUT1 | 15.37 1.78 0.00 | 210 | 0 | 4983 | 9.04 1.27 0.00 | 162 | 0 | 4872 |
| | | FUT2 | 9.21 0.98 0.00 | 139 | 0 | 4476 | 7.32 0.91 0.00 | 170 | 0 | 4575 |
| | 100m | CTRL | 20.25 3.40 0.00 | 213 | 0 | 5112 | 30.51 4.31 0.00 | 213 | 0 | 5112 |
| | | FUT1 | 16.71 2.26 0.00 | 210 | 0 | 4986 | 9.28 1.59 0.00 | 162 | 0 | 4875 |
| | | FUT2 | 10.17 1.30 0.00 | 139 | 0 | 4329 | 8.43 1.20 0.00 | 170 | 0 | 4614 |
| | 200m | CTRL | 21.55 4.93 0.02 | 213 | 0 | 5112 | 33.74 6.75 0.01 | 213 | 0 | 5112 |
| | | FUT1 | 18.02 2.97 0.01 | 210 | 0 | 4992 | 11.08 2.33 0.02 | 162 | 0 | 4866 |
| | | FUT2 | 11.09 1.68 0.00 | 141 | 0 | 4392 | 10.18 1.79 0.00 | 169 | 0 | 4650 |
| | 300m | CTRL | 23.63 6.57 0.07 | 213 | 57 | 5112 | 37.82 9.85 0.04 | 213 | 15 | 5112 |
| | | FUT1 | 20.41 3.99 0.04 | 210 | 0 | 5004 | 13.09 3.06 0.07 | 163 | 48 | 4833 |
| | | FUT2 | 12.22 2.34 0.01 | 142 | 0 | 4434 | 13.81 2.45 0.04 | 169 | 3 | 4749 |
| | 400m | CTRL | 31.06 8.69 0.12 | 213 | 222 | 5112 | 41.90 13.13 0.07 | 213 | 150 | 5112 |
| | | FUT1 | 21.85 5.52 0.08 | 210 | 456 | 5013 | 21.71 4.26 0.16 | 192 | 0 | 4896 |
| | | FUT2 | 14.83 3.19 0.05 | 142 | 66 | 4515 | 18.16 3.62 0.09 | 179 | 105 | 4779 |





**Table A4.** Numerical results for Rutan (area 5, central Sweden). CTRL is for the historical period (1985–2005), FUT1 is the mid-century (2040–2060) and FUT2 the end-of-century (2080–2100). For each time period and GCM (EC-EARTH and GFDL-CM3, respectively), six numerical values are given. IM (Max Avg Min) is for the regional maximum, average and minimum of the 99th quantile of ice mass. IE Max denotes the maximum ice episode duration taken from the average of the area and IH (Min & Max) denote the minimum and maximum icing hour amounts, taken also from the regional average.

| | | | EC-EARTH | | | | GFDL-CM3 | | | |
|---|---|---|---|---|---|---|---|---|---|---|
| **Area** | **Height** | **Year** | **IM** | **IE** | **IH** | **IH** | **IM** | **IE** | **IH** | **IH** |
| | | | Max Avg Min | Max | Min | Max | Max Avg Min | Max | Min | Max |
| 5. | 50m | CTRL | 0.87 0.15 0.01 | 127 | 0 | 3339 | 1.12 0.22 0.02 | 143 | 0 | 3666 |
| | | FUT1 | 0.37 0.10 0.00 | 68 | 0 | 3150 | 0.38 0.09 0.00 | 75 | 0 | 2649 |
| | | FUT2 | 0.32 0.08 0.00 | 78 | 0 | 2328 | 0.14 0.05 0.00 | 38 | 0 | 1560 |
| | 100m | CTRL | 1.07 0.25 0.02 | 131 | 0 | 3609 | 1.34 0.34 0.03 | 134 | 0 | 3660 |
| | | FUT1 | 0.66 0.21 0.02 | 66 | 207 | 3294 | 0.60 0.21 0.02 | 76 | 84 | 2796 |
| | | FUT2 | 1.0 0.39 0.06 | 80 | 33 | 2514 | 0.56 0.23 0.04 | 38 | 9 | 1812 |
| | 200m | CTRL | 2.25 1.20 0.55 | 133 | 51 | 3864 | 2.84 1.08 0.52 | 143 | 0 | 3987 |
| | | FUT1 | 1.15 0.80 0.42 | 120 | 207 | 3387 | 1.14 0.78 0.50 | 105 | 84 | 3441 |
| | | FUT2 | 0.75 0.40 0.20 | 51 | 33 | 2535 | 0.71 0.43 0.24 | 53 | 9 | 2454 |
| | 300m | CTRL | 2.78 1.18 0.25 | 144 | 654 | 4056 | 3.71 1.64 0.36 | 144 | 606 | 4230 |
| | | FUT1 | 2.48 1.07 0.23 | 68 | 384 | 3843 | 2.23 0.83 0.18 | 79 | 99 | 3066 |
| | | FUT2 | 1.64 0.73 0.18 | 80 | 75 | 2754 | 1.05 0.47 0.11 | 38 | 81 | 2229 |
| | 400m | CTRL | 4.25 1.84 0.57 | 149 | 1068 | 4281 | 3.82 2.06 0.65 | 144 | 615 | 4251 |
| | | FUT1 | 3.45 1.67 0.44 | 81 | 492 | 3951 | 2.81 1.25 0.35 | 80 | 0 | 3411 |
| | | FUT2 | 2.78 1.19 0.46 | 89 | 141 | 2997 | 1.54 0.77 0.26 | 38 | 219 | 2304 |



*Author contributions.* Oskari Rockas: Data analysis, Writing (main responsible for draft preparation and editing). Pia Isolähteenmäki: Data analysis, Writing (draft preparation, review). Marko Laine: Supervision, Writing (review, editing), Project administration. Anders J. Lindfors: Supervision, Writing (review), Project administration. Karoliina Hämäläinen: Supervision, Writing (review). Anton Laakso: HCLIM data contact, Writing (review).

*Competing interests.*   The authors declare that they have no conflict of interest.

*Acknowledgements.*   The HCLIM data used in this study has been produced by the NorCP project, which is a Nordic collaboration involving climate modeling groups from the Danish Meteorological Institute (DMI), Finnish Meteorological Institute (FMI), Norwegian Meteorological Institute (MET Norway) and the Swedish Meteorological and Hydrological Institute (SMHI). This research was funded by EU Horizon project RISKADAPT (project number 101093939).



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
