# Peer review of "Future Rime Ice Conditions for Energy Infrastructure over Fennoscandia Resolved with a High-Resolution Regional Climate Model"

_EGUsphere, 2025_

## Referee Comment (RC2)

**Comment on "Future Rime Ice Conditions for Energy Infrastructure over Fennoscandia Resolved with a High-Resolution Regional Climate Model" from Oskari Rockas, Pia Isolähteenmäki, Marko Laine, Anders V. Lindfors, Karoliina Hämäläinen and Anton Laakso**

**General Comment**

With this paper, the authors present an important contribution to the highly relevant question on how atmospheric icing can change regionally in the future under the conditions of climate change. So far, there are still only a few international studies on this problem. The paper shows how high-resolution regional results for Fennoscandia were obtained by the use of an ice accretion model that utilizes the outputs from the high-resolution regional climate model HCLIM driven by results of two global climate models for the RCP 8.5 emission scenario.

The results for rime ice are presented and discussed for two time periods in the future. They show "a general decrease for in-cloud icing conditions" over Fennoscandia for two time periods in the future, compared to the historical period with exceptions in the northern parts of Fennoscandia and locally over higher altitudes, where increasing trends are found. Authors suggest that "the main driver for the decrease of in-cloud icing conditions over Fennoscandia is the warming trend in temperatures. Although the warmer atmosphere allows for a higher moisture content, icing does not occur when temperatures are above zero degrees Celsius", i.e. for most regions of Fennoscandia. They explain exceptions by the "…increasing trend [of in cloud icing] over some of the northern regions in mid-century could be explained by the increase of LWC over regions where freezing temperatures remain, but, on the other hand, the temperatures are not too cold, allowing water to stay in liquid form." This interpretation is consistent with and supports the findings of other authors.

If one evaluates these results, it is clearly evident that more research is needed to analyze future ice accretion conditions in order to provide society and infrastructure operators with robust data on ice accretion on structures in the future. The paper is in accordance with the scope of the journal.

The main concerns relate to certain inconsistencies between the title, abstract and conclusion of the paper and its content, which also reports regional results for 7 area and for specific application fields (power lines and wind turbines). Title says "Future Rime Ice Conditions for Energy Infrastructure over Fennoscandia …", the abstract states "Thus, since climate change is expected to impact winter weather conditions in northern Europe, its effects on atmospheric icing occurrence over the Fennoscandian region …" and the conclusions summarize "The objective of this study was to assess the simulated changes in rime ice formation during in-cloud icing episodes across the Fennoscandian region …" The only "hint" to more specific results is made in the title by "for Energy Infrastructure". The first time that specific evaluations are mentioned is in "2.3 Processing of the ice model outputs" with "The icing model output was further processed to allow a comprehensive evaluation of the changes in icing climate from multiple perspectives. …"

The paper must be revised to consider, whether it needs to take two regional (region-specific) perspectives into account. If so, it needs to be addressed (in more detail) at least in the abstract, in the introduction (Was there any project related information and/or information from power lines or wind parks?) and in the conclusions.

Furthermore, the paper has to be (significantly) improved in two important aspects.

I.  There is no plausibility check (validation is not even required because the availability of icing data is sparse) of the results of the icing model or (almost all) other parameters used in, not even with references to other papers. The absolute values for ice masses in Table 2 and (at least) tables A.1 and A.2 as well as the changes in absolute ice masses in Figure A1 seem to be to small compared to results from other studies (for instance Iversen et al., 2023; even when taking in mind the different measures for extreme values) if one takes in mind that the results are for heights >= 50 m above terrain.

II. There is no information on whether or not a climate change signal is detected. As one can expect a significant natural climate variability even for icing data, it needs to be checked (tested), if the changes in the results of modelled icing data are significant, at least due to the very small absolute values and absolute changes in ice masses (see Figures 4 and A1 for instance).

Keeping the comments and requirements mentioned before into consideration, the paper needs to be re-structured (at least amended). The main topics are mostly clear and understandable. The methods need to be improved in order to get plausible results and reliable conclusions.

**Specific comments**

Structure of comments: Line: "Citation (if necessary)", Comment

1) 33: "… rime ice … loads …" are used by reference to Iversen et al. (2023). This is right.
   55: " … mean and annual maximum ice loads …". This needs to be checked for the whole paper as the term is used alternately with "ice mass". (In order to be precise, "ice load" should refer to a force with unit [N/m], or similar whereas "ice mass" refers to a (unit length related) mass with [kg/m]. Please check the units throughout the paper, too, because sometimes only [kg] is used for (unit length related) mass.

2) 61: Most of the statements regarding the rime ice model refer to the paper by Hämäläinen and Niemelä (2017). One of the Conclusions in this paper is "The model is not able to forecast the accumulated ice mass precisely. However, in an icing atlas type of product, the frequency of ON–OFF behaviour is a more critical parameter than is absolute ice mass." The "Results of the observation comparison" state "The purpose of this case study was to show that the model simulates icing events in a realistic manner. The ice mass can grow and melt by responding to atmospheric forcing correctly. On the other hand, the absolute values are not precise. However, the main goal here is to estimate the length of the icing periods, and therefore, absolute ice mass values have less importance." Please show, how the model has been improved in order to produce more realistic results for ice mass or discuss the topic with respect to the results of this paper later on (this is a key point related to I.). Are ther new results from Luosto? Are there data that can be used from wind parks and/or power lines in areas 1-7?

3) 78: "various parameters related to icing are calculated; most notably the ice mass (gm−1), but also the ice density (gm−3), the total diameter of the cylinder and ice"
   62: "Rime ice accretion is modeled over a vertical (length = 1 m), freely rotating standard cylinder with icing rate (gs−1)"
   The diameter of the "standard cylinder" is not specified. Does "standard cylinder" mean "reference collector" as defined in ISO 12494 (2017), i.e. a diameter of 0.03 m?

4) 79: "Makkonen Lasse, 1984 … and Stallbrass J.R", unify the form of references here and in the references section. Delete first names or initials here (and in references) or add it elsewhere.

5) 115: "Figure 1 displays the specific areas referenced in Table 1" There is a lack of key information regarding the specified areas: Total area of each, number of grid points, relief

information (mean, maximum and minimum altitude, for instance). These are supplementary information for understanding the results of the ice accretion.

6) 118: "a maximum of all grid points was calculated for each time step" Do you mean "the maximum value from all grid points of the area was determined for each time step"?

7) 120: "For areas 1–5, extreme values of ice mass were calculated by taking the 99th quantile of the 20-year periods" What values of ice mass were used how? All ice mass values from all grid points of the area from each time step for the whole period? In Table 2 and A1-to A4 you refer to "IM (Max Avg Min) is for the regional maximum, average and minimum of the 99th quantile of ice mass" That means, you determine the "99th quantile of ice mass for all grid points of the area from each time step for the whole period and determine from that maximum, average and minimum values for the area"? Please clarify.

8) 123: "The icing hours and episode durations were calculated from the average of the grid points in the area." Does it mean "The annual icing hours were calculated from the average of all grid points values in the area, where the respective grid point values fulfill specific icing conditions (defined thereafter) during a year are counted (and adjusted due to the time resolution of 3 hours?) to the annual total sum of icing hours" How do you calculate the annual episode duration of icing? Please specify in more detail.

9) 138: "Changes in ice load (mean and annual maximum)" means "Changes in ice load (annual mean and annual maximum per grid point"?)

10) 148 and the whole description thereafter: "The relative trend is pronounced, with some regions experiencing close to a 100 % decrease, indicating conditions where icing would no longer occur. However, the mean loads in these regions are relatively low to begin with, resulting in a small absolute change". See II.

11) 180 and thereafter: "Both the annual mean and the annual maximum are calculated for the regional maximum of the grid points." See 5) That means, that each box in the plots in Fig. 4 for each time period consist of 20 values (annual mean or annual max for each of the 20 years of the time period)? With reference to Fig. 4 only the trends of median values are discussed (even though the median is mentioned only one time). What about the variability of results, represented by boxes and whiskers. The variability seems to change for different time periods (in different ways). Do we really see "evident" trends for the annual means and a less clear "signal" for annual maxima? See II.

12) Figure 4: "Ice load [kg]" is twice over, see 1).

13) 206: "note that the model output was calculated for a cylinder of diameter of 3 cm" Finally found the diameter of the cylinder! Needs to be defined in 2.1 already, see 3).

14) 245: "(International Organization for Standardization, 2017)", unify the form of references here and in the references section. "ISO 12494" is used before

15) 260: "Some areas in … southernmost Sweden, … may experience close to zero months of freezing conditions at the end-of-century". Does it coincide with results for icing?

16) 263 and thereafter (chapter 3.2.2): Whereas in the chapter for temperature before, the discussion for LWC is more a general one (regarding changes) and does not point to icing conditions. This is done later on but could be here already (as for temperature before).

17) 274 and thereafter (chapter 3.2.3): see 16) but valid here for wind. For instance, it could be discussed, why some regions with increasing wind speed show a decrease in icing and vice versa.

18) 291: "near the surface" What does it mean? >=50 m above ground? Is that near to the surface?

19) 293: ". The mid-century increase is approximately 30–50%, while the strongest negative changes approach 100% toward the end-of-century." This is a misleading formulation. It seems, that "The mid-century increase is approximately 30–50%" refers to "there is a

temporary increasing trend during mid-century … over parts of northern Fennoscandia and at higher altitudes" from the sentence before. But "while the strongest negative changes approach 100% toward the end-of-century" seems to refer to "decreasing trend in mean rime ice conditions near the surface over most of Fennoscandia by the end-of-century." (see Fig 2 a). Consider rephrasing.

20) 312: "is expected"? Do you mean it in the sense of "expectation" or do you mean it in the sense of "has been analyzed" (or similar)?

21) 318: "Correspondingly, this can explain the increase in the annual maxima for the end-of-century; even with an enhanced temperature increase, the likewise increased LWC allows for temporarily large ice loads." Does it mean higher variability, too? How does this match with the results for areas 6 and 7, where variability seems to decrease in area 7 (and increases in area 6).

22) 322: "In our study,the absolute ice loads are smaller compared to the ensemble means in the study of Lutz et al. (2019) and thus may represent the lower end of the estimated ice distribution." This is the conclusion chapter. It should be discussed in 3.1 and can be one contribution to aspect I.

23) 324: "This was supported by our rough comparisons with LWC obtained from the 325 CERRA reanalysis data (stands for Copernicus European Regional Reanalysis, (Ridal et al., 2024)). For 50 and 400 meters, about two times smaller amounts of mean LWC were observed in Finland in HCLIM data (not shown); however, for monthly maxima, HCLIM data showed larger amounts." See (put it to) 16) as a contribution to aspect I.

24) 328: "Moreover, atmospheric icing encompasses various subtypes, each with distinct formation processes, which rely differently on influencing parameters such as ice density. The impact and severity of different icing types vary significantly depending on the affected target: aviation and in-cloud icing, forests and wet-snow-damages, passive-icing episodes and wind turbines, and ice loads over power lines or electric towers." These statements are right but what do the contribute to the conclusions with respect to all the analysis before? Are the results only being useful for wind turbines and powerlines and not for aviation and forests? Why this is needed to be emphasized because the area specific analyses point to the wind power and power lines.

---

## Author Comment (AC1)

*Response to comments by Reviewer #1 concerning our manuscript "Future Rime Ice Conditions for Energy Infrastructure over Fennoscandia Resolved with a High-Resolution Regional Climate Model" by Oskari Rockas et al.*

We thank the reviewer for their positive and constructive comments, which have helped us to improve our manuscript. Below, we detail our responses to the received comments.

**Major point**

*Please double check the absolute units/amount you get for ice masses. The values you show for instance in Figure 4a) and 4c), Table 2 and Figure A1 are very low compared to e.g. Iversen et al. 2023, considering that an ice load < 1kg would not be an issue I guess and that a max icing thickness of 70 mm is shown in Fig. 4d). Also, I think in Fig 4a) and 4c), the unit should be [kg/m] not [kg].*

Reply: The units and amounts have been double-checked and, for example, in table A3 quite large maximum ice masses can be seen across all heights and models (more than 20 kg/m). The location there is partly in the Scandinavian mountains, which explains the large ice masses, but also illustrates that from HCLIM input, larger amounts can be produced. However, the major point you raised is also noted in the paper: "the absolute ice loads are smaller compared to the ensemble means in the study of Lutz et al. (2019) and thus may represent the lower end of the estimated ice distribution". We contribute this to uncertainty in liquid water content in HCLIM simulations, and LWC has been recognized as the largest source of uncertainty in icing frequency, for example by Lutz et al. (2019). In our response to Reviewer #2 (Aspect I), we have a wider answer to this point and how we plan to address it in the revised version of the article.

**Abstract**

l6: two twenty-year periods -> two *future* twenty-year periods

Reply: Very true, future should be highlighted in this case, and it will be fixed accordingly.

**1. Introduction**

l54: expense of small ensemble size. -> expense of *a* small ensemble size *and usually shorter time periods*.

Reply: Yes, these are good grammatical and substantive corrections, will be added to the text.

**2.1 Rime ice model**

l68: close to 1 for large particles (for which inertia dominates) and vice versa for small particles -> close to 1 for large particles (for which inertia dominates) and *close to 0 for small particles*.

Reply: Your suggestion is a clearer explanation so this will be fixed as suggested.

**2.2 HARMONIE-Climate**

l96: EC-EARTH shows a colder and drier response to climate change in northern Europe compared to GFDL-CM3. -> May be confusing or interpreted as EC-EARTH showing a cooling and drying. Maybe change to "EC-EARTH shows a more moderate warming and increasing humidity response to climate change in northern Europe than to GFDL-CM3."
l103: I would drop the "business as usual" remark for RCP85. It is a debated term, and it does not add any relevant information.
l104: A similar remark; whether "RCP 8.5 is considered to be increasingly unlikely" is also disputed, I'd say. I don't think there is a common agreement on this. Especially taking recent global developments into account. -> "Although RCP 8.5 may be considered to be increasingly unlikely ..."
l105: "it remains to be an useful tool for" -> "it remains *a useful tool* for"

Reply: For l96, true, this wording can be a bit confusing. It will be changed to something akin what you suggested. For l103, true, it doesn't add important information so it will be removed. For l104, that is a good correction so this will be changed according to your correction and the grammatical error in l105 will be fixed.

**2.3 Processing of the ice model outputs**

l109: 1) results calculated for 50 meters -> 1) results calculated for 50 meters *above ground*
l117: calculated from all heights -> calculated *for* all heights
l118: a maximum of all grid points -> *the maximum of all grid points*

Reply: For l109, yes, this correction makes sense and will be added to the final text. The grammatical corrections in l117 and l118 will be fixed as suggested.

**3.1.1 Fennoscandia (50 m) -> Fennoscandia (50 m *height*)**

l148: the GFDL-CM3 boundary model -> the GFDL-CM3 boundary *data*

Reply: This will be corrected as suggested as well as the header of the section.

**3.1.2 Test areas 6–7: Power line perspective (50m)**

l176-180: "In box plots ..." -> You may consider dropping the description of the figure (e.g. "solid lines" etc.) in the text and leave it to the caption of the figure.
l187: both model configurations show increase -> both model configurations show *an* increase
l192: "Yellow represents the historical period, red the mid-century, and black the end-of-century." -> Again, consider dropping the description of the figure in the text and leave it to the caption of the figure.

Reply: For l176-180: Yes, we agree that the description of the image can be taken out of the main text and so it will be corrected as suggested. The grammatical error in l187 will be fixed

as suggested. For l192: The same applies here as in the previous comment about image descriptions in the text, it will be corrected as suggested.

**3.1.3 Test areas 1-5: Wind power perspective (50-400m)**

l209: The mean value is highlighted with a diamond symbol. -> Again, consider dropping the description of the figure in the text and leave it to the caption of the figure.

Reply: Yes, the description will be removed from the main text.

**3.2.1 Temperature**

l249: In Fig. A3 -> In case you have not reached a figure limit, move this to the main text. It seems relevant information for the main text to me.

Reply: The figure can be moved to the main text, we have not reached a figure limit.

**3.2.2 Liquid water content**

l265: "LWC only included the cloud liquid water content (CLWC) because cloud rain water content (CRWC) was not available in the pre-calculated HCLIM data. Thus, the modeled atmospheric ice type is primarily rime ice." -> I don't entirely understand what you refer to with cloud rain water content (CRWC) and how it relates to total LWC. Please explain in more details (maybe refering to relevant literature) and maybe move this part to section 2.

We agree that this notion was not opened enough in the preprint version of the article and will be reworked and clarified in the revised version. A better explanation, and which can be taken to section 2 as suggested, is that liquid water content in an atmospheric layer can be divided into cloud liquid water content (CLWC) and rain liquid water content (RLWC), so into nonprecipitating cloud droplets and rain droplets (Tian et al. 2019, Ellis and Vivekanandan 2011). HCLIM model bases its cloud microphysics in an ICE3-OCND2-scheme which separates between cloud water and rain (Lind et al. 2020, Bengtsson et al. 2017), however, the simulations we used as input data in our research only included the cloud liquid water content (CLWC) and not the rain liquid water content (RLWC).

**3.2.3 Wind speed**

l275: where the possible ice forms. -> "where ice may form." or "where it may turn into ice."

These are better suggestions, it will most probably be changed to "where ice may form" as suggested.

**4 Conclusions**

l333: "is a business-as-usual scenario" -> (maybe) consider changing it to a more neutral term, e.g. "is a high emission scenario"

Reply: Yes, as already discussed, the "business-as-usual" can be avoided, and, in this case, be changed into "is a high emission scenario" as suggested.

**Table 1:** the domains to which -> the domains *for* which

Reply: This will be corrected as suggested.

**Table 2 and A1- A4:** Please add the units of IM, IE and IH to the table. Either in the table or the caption.

Reply: The units will be added.

Bengtsson, L., Andrae, U., Aspelien, T., Batrak, Y., Calvo, J., de Rooy, W., Gleeson, E., Hansen-Sass, B., Homleid, M., Hortal, M., et al.: The HARMONIE‑AROME model configuration in the ALADIN‑HIRLAM NWP system, Monthly Weather Review, 145, 1919‑1935, 2017.

Ellis, Scott M., and Jothiram Vivekanandan. "Liquid water content estimates using simultaneous S and K a band radar measurements." *Radio Science* 46.02 (2011): 1-15.

Lind, P., Belušic, D., Christensen, O. B., Dobler, A., Kjellstrom, E., Landgren, O., Lindstedt, D., Matte, D., Pedersen, R. A., Toivonen,E., and Wang, F.: Benefits and added value of convection-permitting climate modeling over Fenno-Scandinavia, Climate Dynamics, 55,1893‑1912, https://doi.org/10.1007/s00382-020-05359-3, 2020.

Lutz, J., Dobler, A., Nygaard, B. E., Mc Innes, H., and Haugen, J. E.: Future projections of icing on power lines over Norway, in: Proceedings of the International Workshop on Atmospheric Icing of Structures IWAIS, 2019.

Tian, Jingjing, et al. "Estimation of liquid water path below the melting layer in stratiform precipitation systems using radar measurements during MC3E." *Atmospheric Measurement Techniques* 12.7 (2019): 3743-3759.

---

## Author Comment (AC2)

*Response to comments by Reviewer #2 concerning our manuscript "Future Rime Ice Conditions for Energy Infrastructure over Fennoscandia Resolved with a High-Resolution Regional Climate Model" by Oskari Rockas et al.*

We thank the reviewer for their constructive comments, which have helped us to improve our manuscript. Below, we detail our responses to the received comments.

*Answer to the general comments*

*"The paper must be revised to consider, whether it needs to take two regional (region-specific) perspectives into account. If so, it needs to be addressed (in more detail) at least in the abstract, in the introduction (Was there any project related information and/or information from power lines or wind parks?) and in the conclusions."*

The concern that there is little mention of the area-specific results outside the Results section will be addressed in the revised version of the article. To the abstract, the sentence "The results suggest a general decrease in in-cloud icing conditions over northern Europe compared to the historical period (1985–2005)." will be changed to "The results, including results both from the whole Fennoscandian area and from specific regions relating to energy infrastructure suggest a general decrease in in-cloud icing conditions over northern Europe compared to the historical period (1985–2005).". Similar mentions will be added to the end of the Introduction chapter, around rows 49-51 and 55, and the first sentence of the Conclusions chapter will also be reworked to emphasize the area-specific results.

We see value in the region-specific approach as it allows us to show details that are not seen in the general Fennoscandian results, such as the relationship between the icing episode length and the maximum thickness obtained in the episode (and the difference in the two locations). Moreover, area 6 was specifically researched in the RISKADAPT-project under which the research has been conducted; area 7 was chosen to represent different icing conditions while still coinciding with power transmission lines in Finland. As said in the paper, areas 1-5 "were selected to coincide with existing or planned wind parks", but they are not linked to the RISKADAPT-project per se.

**Aspect I:** *"There is no plausibility check (validation is not even required because the availability of icing data is sparse) of the results of the icing model or (almost all) other parameters used in, not even with references to other papers. The absolute values for ice masses in Table 2 and (at least) tables A.1 and A.2 as well as the changes in absolute ice masses in Figure A1 seem to be to small compared to results from other studies (for instance Iversen et al., 2023; even when taking in mind the different measures for extreme values) if one takes in mind that the results are for heights >= 50 m above terrain."*

Thank you for highlighting this important point. We agree that a plausibility check is essential. While direct validation of icing is challenging due to sparse observational data,

we have examined the plausibility of the input climate parameters and compared our results with previous studies. Next, we summarize these aspects:

*Forcing data*: Among the three parameters fed to the icing model, temperature (from 2 meters) is the most thoroughly validated. Lind et al. (2022), using the same HCLIM simulations as our study, report small winter biases (median: about –0.5 °C for EC-EARTH and +1 °C for GFDL-CM3), with seasonal biases generally within ±2 °C. Furthermore, nearly all changes in daily maximum and minimum temperatures are statistically significant.

For wind speed, changes are less robust. Lind et al. (2022) and related studies (Christensen et al., 2022; Kjellström et al., 2011; Tobin et al., 2016) indicate that future changes in 10 m wind speed over northern Europe are associated with large uncertainties and weak signals. We will emphasize this in the revised manuscript.

In contrast, LWC has not been evaluated at all for HCLIM in the winter months, making it a large source of uncertainty in our icing estimates. As noted already in the article (relating to comment no. 22), previous research conducted by Lutz et al. (2019) showed that differences in LWC between RCMs can lead to large variations in ice loads – ranging from 0-25 kg/m to 25-250 kg/m in a mountainous grid point depending on the driving RCM, which they attributed to differences in mean LWC between the RCMs (0.08 g/kg vs. 0.15 g/kg). In general, notes on LWC amounts being the largest uncertainty with icing are also referenced in both Hämäläinen & Niemelä (2017), and Hämäläinen et al. (2020). Additionally, our rough comparison with CERRA reanalysis (in reference to comment no. 23), shows the mean CLW in HCLIM (EC-EARTH) in 400 meters being a closer match with mean CERRA CLW in 50 meters than with CERRA CLW in 400 meters (see Figure 1, found in the end of the document). However, the strongly increasing change signal in LWC is in accordance with estimated moister winters projected for northern Europe (Ruosteenoja and Räisänen, 2013).

*Previous studies*: As seen from the example in Lutz et al. (2019), there exists large variation in the absolute icing estimates. Iversen et al. (2023) saw 10-year rime ice loads of 0-5/10 kg/m in lower altitudes and up to at least 30 kg/m in the Scandinavian mountains while we saw 99th quantile loads of 0-5 kg/m (when taking all heights into consideration) in lower altitudes and up to at least 20-40 kg/m in the Scandinavian mountains.

*Sensitivity test*: To explore potential underestimation, we ran a sensitivity test using LWC from 400 m for a Finnish area at 50 m height (EC-EARTH, historical and end-of-century periods, shown in table 1 in this response for areas 1-3 referenced in the actual article). Results show increased icing amounts in the historical period, but strong warming still limits icing in the future period. We acknowledge that this test is not physically

consistent, as LWC and temperature are not independent, but it illustrates the impact of LWC uncertainty.

Taking all this into account, we will:

- Add a section at the start of the Results chapter discussing climate data plausibility and uncertainties, highlighting LWC as the dominant source.
- Also discuss uncertainties in the icing model itself (as noted in comment no. 2).
- Compare our results explicitly with Iversen et al. (2023) and Lutz et al. (2019), emphasizing variability and that our estimates are on the lower end, at least in lower elevations and lower heights.
- Clearly state that the absolute values carry uncertainty but are shown transparently to inform future research.

**Aspect II:** *"There is no information on whether or not a climate change signal is detected. As one can expect a significant natural climate variability even for icing data, it needs to be checked (tested), if the changes in the results of modelled icing data are significant, at least due to the very small absolute values and absolute changes in ice masses (see Figures 4 and A1 for instance)."*

True, there should be more done to address whether there is an actual signal or not, especially with the problem that arose in Aspect I. We will be addressing this problem in two ways: first, by following the example of Iversen et al. (2023) (while they followed for example Kendon et al. 2008) and performing spatial averaging to better separate a climate change signal from the data as it reduces grid-box noise.

Secondly, we will be using a two-sample t-test to look for statistical significance in our trend results (annual mean ice load, annual maximum ice load and annual mean icing hours) with a 0.1 confidence level, originally. However, when a statistical test is performed for a large set of grid points, this can mean that many of the points that seem to be statistically significant are not. Thus, we will adjust the 0.1 level by calculating a false discovery rate for each set of significance tests by following Wilks, 2016. An example of this can be seen in Figure 3 in this response, which is a test version of the annual maximum ice load calculations for the 50-meter height with only statistically significant (adjusted by false discovery rate) results shown; what stands out is the increase in northern Lapland across the GCMs and time periods and the decrease in GFDL-CM3 especially for end-of-century. In the revised version of the article, the significant areas will rather be highlighted with hatching.

*Answers to the specific comments*

**1)** 33: "… rime ice … loads …" are used by reference to Iversen et al. (2023). This is right.
55: " … mean and annual maximum ice loads …". This needs to be checked for the whole paper as the term is used alternately with "ice mass". (In order to be precise, "ice load" should refer to a force with unit [N/m], or similar whereas "ice mass" refers to a (unit length related) mass with [kg/m]. Please check the units throughout the paper, too, because sometimes only [kg] is used for (unit length related) mass.

Reply: "Ice load" will be changed to "ice mass" in most of the paper according to your comment, as [kg/m] is what our ice model produces.

**2)** 61: Most of the statements regarding the rime ice model refer to the paper by Hämäläinen and Niemelä (2017). One of the Conclusions in this paper is "The model is not able to forecast the accumulated ice mass precisely. However, in an icing atlas type of product, the frequency of ON–OFF behaviour is a more critical parameter than is absolute ice mass." The "Results of the observation comparison" state "The purpose of this case study was to show that the model simulates icing events in a realistic manner. The ice mass can grow and melt by responding to atmospheric forcing correctly. On the other hand, the absolute values are not precise. However, the main goal here is to estimate the length of the icing periods, and therefore, absolute ice mass values have less importance." Please show, how the model has been improved in order to produce more realistic results for ice mass or discuss the topic with respect to the results of this paper later on (this is a key point related to I.). Are ther new results from Luosto? Are there data that can be used from wind parks and/or power lines in areas 1-7?

Reply: The model is the same as in Hämäläinen & Niemelä (2017), so as you note, the icing model's skill can be a contribution to discuss the problem with absolute values. In Luosto, icing observations are no longer conducted (instead, Vehmasmäki has continuing, ON/OFF-type observations), and, unfortunately, we have no knowledge of icing observations from areas 1-7; for example, wind parks are privately owned in Finland, and their data is not freely available. However, the icing model is run operatively nowadays at FMI with weather information from the regional weather model used in Finland (MEPS, HARMONIE-AROME based). It has been verified both against the icing and ceilometer observations in Vehmasmäki (not in article form) and found good accordance. An example of the ceilometer verification can be seen close to the end of this response (Figure 2). Already before, Hämäläinen et al. (2020) verified the icing model against multiple ceilometer profiles and found a mostly good match between observations and modelled icing in inland stations and less so near coastal areas at heights below 400 meters.

**3)** 78: "various parameters related to icing are calculated; most notably the ice mass (gm−1), but also the ice density (gm−3), the total diameter of the cylinder and ice"
62: "Rime ice accretion is modeled over a vertical (length = 1 m), freely rotating standard cylinder with icing rate (gs−1)"
The diameter of the "standard cylinder" is not specified. Does "standard cylinder" mean "reference collector" as defined in ISO 12494 (2017), i.e. a diameter of 0.03 m?

**13)** 206: "note that the model output was calculated for a cylinder of diameter of 3 cm" Finally

found the diameter of the cylinder! Needs to be defined in 2.1 already, see 3)

Reply: As noted in comment no. 13, the diameter of the cylinder is 3 cm, referencing the "reference collector" defined by the icing standard (International Organization for Standardization, 2017). This will be clarified already in chapter 2.1 as suggested.

**4)** 79: "Makkonen Lasse, 1984 … and Stallbrass J.R", unify the form of references here and in the references section. Delete first names or initials here (and in references) or add it elsewhere.

Reply: References are corrected here so that first names and initials are removed from the reference in row 79 to unify the form with the rest of the article.

**5)** 115: "Figure 1 displays the specific areas referenced in Table 1" There is a lack of key information regarding the specified areas: Total area of each, number of grid points, relief information (mean, maximum and minimum altitude, for instance). These are supplementary information for understanding the results of the ice accretion.

Reply: This supplementary information on the areas will be added as a separate table in the Appendix section.

**6)** 118: "a maximum of all grid points was calculated for each time step" Do you mean "the maximum value from all grid points of the area was determined for each time step"?

Reply: Yes, the case is as noted in your comment. This information will be added to the description.

**7)** 120: "For areas 1–5, extreme values of ice mass were calculated by taking the 99th quantile of the 20-year periods" What values of ice mass were used how? All ice mass values from all grid points of the area from each time step for the whole period? In Table 2 and A1-to A4 you refer to "IM (Max Avg Min) is for the regional maximum, average and minimum of the 99th quantile of ice mass" That means, you determine the "99th quantile of ice mass for all grid points of the area from each time step for the whole period and determine from that maximum, average and minimum values for the area"? Please clarify.

Reply: The 99th quantile of ice mass is determined for each grid point of the area in a 20-year period. Then, from those 99th quantile values, the regional maximum, average, and minimum values are calculated. This will be clarified in the text and the tables.

**8)** 123: "The icing hours and episode durations were calculated from the average of the grid points in the area." Does it mean "The annual icing hours were calculated from the average of all grid points values in the area, where the respective grid point values fulfill specific icing conditions (defined thereafter) during a year are counted (and adjusted due to the time resolution of 3 hours?) to the annual total sum of icing hours" How do you calculate the annual episode duration of icing? Please specify in more detail.

Reply: First, the average value of ice mass was calculated from all the grid points in the area. Then, it was checked how often this regional average value fulfills specific icing conditions (more than 10 grams of ice present in the cylinder) which gives the annual total sum of icing hours, adjusted due to the time resolution of 3 hours. An icing episode is the number of days there is more than 10 grams of ice present in the cylinder

continuously, also calculated from the regional average value of ice mass. The maximum episode length inside the 20-year period is determined from all icing episode lengths.

**9)** 138: "Changes in ice load (mean and annual maximum)" means "Changes in ice load (annual mean and annual maximum per grid point"?)

Reply: Yes, though in the revised version of the article it will be for a spatial average, as discussed in Aspect II.

**10)** 148 and the whole description thereafter: "The relative trend is pronounced, with some regions experiencing close to a 100 % decrease, indicating conditions where icing would no longer occur. However, the mean loads in these regions are relatively low to begin with, resulting in a small absolute change". See II.

Reply: Yes, and as answered in Aspect II, this will be supported by a check for statistical significance. I think this sort of formulation is important here as this type of strong decrease (even with a small absolute change) can be statistically significant. It could be better to say that "icing would be minimal" as it's not an exactly 100 % decrease. Also good to note (and can be written later in the main text) that the temperature increase is so strong towards the 2081-2100 period in RCP8.5 scenario that the decrease is anyways strong; for example, even when using LWC from 400 m at 50 m, southern areas still have very little icing in the end-of-century period (see the attached table).

**11)** 180 and thereafter: "Both the annual mean and the annual maximum are calculated for the regional maximum of the grid points." See 5) That means, that each box in the plots in Fig. 4 for each time period consist of 20 values (annual mean or annual max for each of the 20 years of the time period)? With reference to Fig. 4 only the trends of median values are discussed (even though the median is mentioned only one time). What about the variability of results, represented by boxes and whiskers. The variability seems to change for different time periods (in different ways). Do we really see "evident" trends for the annual means and a less clear "signal" for annual maxima? See II.

**21)** 318: "Correspondingly, this can explain the increase in the annual maxima for the end-of-century; even with an enhanced temperature increase, the likewise increased LWC allows for temporarily large ice loads." Does it mean higher variability, too? How does this match with the results for areas 6 and 7, where variability seems to decrease in area 7 (and increases in area 6).

Reply: Thank you for these comments. We would still argue that there is also some discussion on the trends of the extremes of the distributions in lines 182-185, however, it can be expanded and reworded. For example, it can be seen that the distribution minima (whiskers) of the annual maxima in area 6 decrease towards the end-of-century – probably linked to the overall little freezing conditions experienced in this area by 2081-2100. Meanwhile, even the distribution minima see some increase in area 7, which is probably linked to the notion that enough freezing conditions remain in northern Lapland alongside with an increased LWC. This is partly speculative, and we feel that further inspection into the variability of the results is outside the scope of our article. Going back

to comment 11, the word "evident" is maybe too strong a phrase here, especially when discussing the results in area 7 (Utsjoki), and thus it will be removed.

**12)** Figure 4: "Ice load [kg]" is twice over, see 1).

Reply: The units will be fixed wherever [kg] is used instead of [kg/m], as that is the correct unit.

**14)** 245: "(International Organization for Standardization, 2017)", unify the form of references here and in the references section. "ISO 12494" is used before.

Reply: This is a good point. We wish to use this form of reference (International Organization for Standardization, 2017) so the previous in-text citation will be transformed into something akin to: "The accretion coefficient α3 differs for dry growth cases (where α3=1) and wet growth cases (where there is a liquid layer on top of the ice surface; α3 < 1). The international standard explains in more detail how these wet growth scenarios are managed in accordance with the accretion coefficient (International Organization for Standardization, 2017)."

**15)** 260: "Some areas in … southernmost Sweden, … may experience close to zero months of freezing conditions at the end-of-century". Does it coincide with results for icing?

Reply: Yes. As can be seen from the results, these are areas with large relative changes when it comes to, for example, mean ice mass and mean annual icing hours, while the absolute changes are very small. With rare freezing conditions, the mean values will experience a large relative drop. This is linked with what is discussed in the comment no. 10, and also, in aspects I and II; with this strong of a general temperature decrease, icing sees a statistically significant decrease.

**16)** 263 and thereafter (chapter 3.2.2): Whereas in the chapter for temperature before, the discussion for LWC is more a general one (regarding changes) and does not point to icing conditions. This is done later on but could be here already (as for temperature before).

Reply: Notes on how the changes in LWC inform icing will be added to chapter 3.2.2 as you have suggested. Also, as suggested in comment no. 22, the information given previously in Conclusions about LWC comparisons between CERRA and HCLIM data will be discussed in chapter 3.2.2 and later referenced in chapter 4. Also, see our answer for aspect I.

**17)** 274 and thereafter (chapter 3.2.3): see 16) but valid here for wind. For instance, it could be discussed, why some regions with increasing wind speed show a decrease in icing and vice versa.

Reply: Yes, the changes in wind speed should be connected to icing more than is written now. Something akin to the following will be added there: "HCLIM with GFDL-CM3 projects an increase of about 10-20 % in the southern areas of the domain in end-of-century. An increase in the wind speed would support an increase in icing amount and hours; however, a decrease is analyzed in these areas. This suggests that the decreasing

time spent under freezing conditions dominates the effects on icing, as CLWC amounts in southern areas of the domain show mostly minor trends."

**18)** 291: "near the surface" What does it mean? >=50 m above ground? Is that near to the surface?

Reply: "Near the surface" is meant to describe the height of 50 meters, the height of which results are presented for the whole Fennoscandian area. You are correct to point out that it is a bit vague term here, and it would be better to use the actual height instead in the sentence.

**19)** 293: ". The mid-century increase is approximately 30–50%, while the strongest negative changes approach 100% toward the end-of-century." This is a misleading formulation. It seems, that "The mid-century increase is approximately 30–50%" refers to "there is a temporary increasing trend during mid-century … over parts of northern Fennoscandia and at higher altitudes" from the sentence before. But "while the strongest negative changes approach 100% toward the end-of-century" seems to refer to "decreasing trend in mean rime ice conditions near the surface over most of Fennoscandia by the end-of-century." (see Fig 2 a). Consider rephrasing.

Reply: Yes, you are correct that the formulation is misleading. This will be reworked so that it is clear that the strongest decrease does not correspond with parts of northern Fennoscandia and higher altitudes, which are projected to experience temporary increase in the icing conditions.

**20)** 312: "is expected"? Do you mean it in the sense of "expectation" or do you mean it in the sense of "has been analyzed" (or similar)?

Reply: "Expected" was meant here as expected based on the results obtained in this research. It was maybe an oversight from our end, as with climate change information, "expected" is too strong a word. For example, "projected" could be a better alternative and is often used in the paper.

**22)** 322: "In our study,the absolute ice loads are smaller compared to the ensemble means in thestudy of Lutz et al. (2019) and thus may represent the lower end of the estimated ice distribution." This is the conclusion chapter. It should be discussed in 3.1 and can be one contribution to aspect I.

Reply: You are correct, this will be addressed in chapter 3.1 as it is an important framework to understand the results better. Also, see our answer to aspect I.

**23)** 324: "This was supported by our rough comparisons with LWC obtained from the 325 CERRA reanalysis data (stands for Copernicus European Regional Reanalysis, (Ridal et al., 2024)). For 50 and 400 meters, about two times smaller amounts of mean LWC were observed in Finland in HCLIM data (not shown); however, for monthly maxima, HCLIM data showed larger amounts." See (put it to) 16) as a contribution to aspect I.

Reply: As noted above in our answer to comment no. 16, this will be discussed in chapter 3.2.2.

**24)** 328: "Moreover, atmospheric icing encompasses various subtypes, each with distinct

formation processes, which rely differently on influencing parameters such as ice density. The impact and severity of different icing types vary significantly depending on the affected target: aviation and in-cloud icing, forests and wet-snow-damages, passive-icing episodes and wind turbines, and ice loads over power lines or electric towers." These statements are right but what do the contribute to the conclusions with respect to all the analysis before? Are the results only being useful for wind turbines and powerlines and not for aviation and forests? Why this is needed to be emphasized because the area specific analyses point to the wind power and power lines.

Reply: You are correct to pinpoint this section of the Conclusions chapter. The two paragraphs between rows 320 and 331 (with text referring to comments 22 and 23) will be reworked in the revised version. In an older version of the article, these sentences made more sense, but they do not contribute to the conclusions in the current format.

[Figure]

Figure 1: Mean CLW (g/kg) in a) 50m in CERRA (1986-89), b) 50 m in HCLIM (EC-EARTH) (1986-89) c) 400 m in CERRA (1986-89), d) 400 m in HCLIM (EC-EARTH) (1986-89)

[Figure]

Figure 2: Icing from a) ceilometer data (sees icing only in the base of the cloud) and b) the icing model from one day (24 March 2018) in Vehmasmäki.

[Figure]

Figure 3: Relative change in annual maximum ice load in HCLIM simulations at 50-meter height with only statistically significant results shown (adjusted by false discovery rate).

a)

| Height | Year | IM Max Avg Min | IE Max | IH Min | IH Max |
|---|---|---|---|---|---|
| 50 m | CTRL | 0.07 0.06 0.04 | 53 | 0 | 1914 |
|  | FUT1 | 0.05 0.04 0.03 | 37 | 0 | 1239 |
|  | FUT2 | 0.03 0.02 0.01 | 15 | 0 | 639 |
| 50 m * | CTRL | 0.23 0.08 0.04 | 57 | 12 | 1791 |
|  | FUT1 |  |  |  |  |
|  | FUT2 | 0.04 0.02 0.01 | 25 | 0 | 618 |

b)

| Height | Year | IM Max Avg Min | IE Max | IH Min | IH Max |
|---|---|---|---|---|---|
| 50 m | CTRL | 0.14 0.09 0.05 | 84 | 0 | 2373 |
|  | FUT1 | 0.07 0.07 0.04 | 54 | 0 | 1881 |
|  | FUT2 | 0.05 0.04 0.02 | 35 | 0 | 1569 |
| 50 m * | CTRL | 0.68 0.46 0.30 | 125 | 612 | 2883 |
|  | FUT1 |  |  |  |  |
|  | FUT2 | 0.13 0.09 0.05 | 39 | 21 | 1509 |

c)

| Height | Year | IM Max Avg Min | IE Max | IH Min | IH Max |
|---|---|---|---|---|---|
| 50 m | CTRL | 0.47 0.22 0.06 | 122 | 75 | 3783 |
|  | FUT1 | 0.76 0.40 0.09 | 141 | 18 | 2958 |
|  | FUT2 | 0.21 0.10 0.04 | 50 | 0 | 1965 |
| 50 m * | CTRL | 1.85 1.12 0.81 | 163 | 2541 | 4119 |
|  | FUT1 |  |  |  |  |
|  | FUT2 | 0.47 0.27 0.21 | 53 | 606 | 2199 |

Table 1: Part of the tabular results (EC-EARTH and 50 m) from a) area 1, b) area 2 and c) area 3 are shown. The 50 m * results are from a test where CLW data from 400 meters was used; only historical and end-of-century periods have been calculated.

Christensen OB, Kjellström E, Dieterich C et al (2022) Atmospheric regional climate projections for the Baltic sea region until 2100. Earth Syst Dyn 13(1):133–157. https:// doi. org/ 10. 5194/esd- 13- 133- 2022

Hämälainen, K. and Niemela, S.: Production of a Numerical Icing Atlas for Finland, Wind Energy, 20, 171‑189, https://doi.org/10.1002/we.1998, 2017.

Hämäläinen, K., Hirsikko, A., Leskinen, A., Komppula, M., O'Connor, E. J., & Niemelä, S. (2020). Evaluating atmospheric icing forecasts with ground-based ceilometer profiles. *Meteorological Applications*, *27*(6), e1964.

Iversen, E. C., Nygaard, B. E., Hodnebrog, O., Sand, M., and Ingvaldsen, K.: Future projections of atmospheric icing in Norway, Cold Regions Science and Technology, 210, https://doi.org/10.1016/j.coldregions.2023.103836, 2023.

Kendon, E.J., Rowell, D.P., Jones, R.G., Buonomo, E., 2008. Robustness of future changes in local precipitation extremes. J. Clim. 21 (17), 4280–4297. https://doi.org/10.1175/2008jcli2082.1.

Kjellström E, Nikulin G, Hansson U et al (2011) 21st century changes in the European climate: uncertainties derived from an ensemble of regional climate model simulations. Tellus A Dyn Meteorol Oceanogr 63(1):24. https:// doi. org/ 10. 1111/j. 1600- 0870. 2010.00475.x

Lind, P., Beluši´c, D., Médus, E., Dobler, A., Pedersen, R. A., Wang, F., Matte, D., Kjellström, E., Landgren, O., Lindstedt, D., Christensen, O. B., and Christensen, J. H.: Climate change information over Fenno-Scandinavia produced with a convection-permitting climate model, Climate Dynamics, https://doi.org/10.1007/s00382-022-06589-3, 2022.

Lutz, J., Dobler, A., Nygaard, B. E., Mc Innes, H., and Haugen, J. E.: Future projections of icing on power lines over Norway, in: Proceedings of the International Workshop on Atmospheric Icing of Structures IWAIS, 2019.

Ridal, M., Bazile, E., Le Moigne, P., Randriamampianina, R., Schimanke, S., Andrae, U., Berggren, L., Brousseau, P., Dahlgren, P., Edvinsson, L., El-Said, A., Glinton, M., Hagelin, S., Hopsch, S., Isaksson, L., Medeiros, P., Olsson, E., Unden, P., and Wang, Z. Q.: CERRA, the Copernicus European Regional Reanalysis system, Quarterly Journal of the Royal Meteorological Society, 150, 3385‑3411, https://doi.org/10.1002/QJ.4764, 2024.

Ruosteenoja, K., & Räisänen, P. (2013). Seasonal changes in solar radiation and relative humidity in Europe in response to global warming. *Journal of Climate*, *26*(8), 2467-2481.

Tobin I, Jerez S, Vautard R et al (2016) Climate change impacts on the power generation potential of a European mid-century wind farms scenario. Environ Res Lett 11(3):034013. https:// doi. org/ 10. 1088/1748- 9326/ 11/3/ 034013

Wilks, D. (2016). "The stippling shows statistically significant grid points": How research results are routinely overstated and overinterpreted, and what to do about it. *Bulletin of the American Meteorological Society*, *97*(12), 2263-2273.